# Field Matching: an Electrostatic Paradigm to Generate and Transfer Data

**Alexander Kolesov** [* 1 2]  **Stepan I. Manukhov** [* 1 3]  **Vladimir V. Palyulin** [1]  **Alexander Korotin** [1 2]

## Abstract

We propose Electrostatic Field Matching (EFM), a novel method that is suitable for both generative modeling and distribution transfer tasks. Our approach is inspired by the physics of an electrical capacitor. We place source and target distributions on the capacitor plates and assign them positive and negative charges, respectively. Then we learn the electrostatic field of the capacitor using a neural network approximator. To map the distributions to each other, we start at one plate of the capacitor and move the samples along the learned electrostatic field lines until they reach the other plate. We theoretically justify that this approach provably yields the distribution transfer. In practice, we demonstrate the performance of our EFM in toy and image data experiments. Our code is available at https://github.com/justkolesov/FieldMatching

## 1. Introduction

The basic task of generative modeling is to learn a transformation between two distributions accessible by i.i.d. samples. The typical scenarios considered are **noise-to-data** (Goodfellow et al., 2014) and **data-to-data** (Zhu et al., 2017). These are usually referred to as the unconditional data generation and data translation, respectively.

Physics is often at the heart of the principles of generative modeling. One of the first attempts to link generative models and physics was made in Energy-Based models (LeCun & Huang, 2005, EBM). They parameterize data distributions using the Gibbs-Boltzmann distribution density and generate data through simulation of Langevin dynamics (Du & Mordatch, 2019; Song & Kingma, 2021).

---

*Equal contribution [1]Skolkovo Institute of Science and Technology, Moscow, Russia [2]Artificial Intelligence Research Institute, Moscow, Russia [3]Lomonosov Moscow State University, Faculty of Physics, Moscow, Russia. Correspondence to: Alexander Kolesov <a.kolesov@skoltech.ru>.

*Proceedings of the $42^{nd}$ International Conference on Machine Learning*, Vancouver, Canada. PMLR 267, 2025. Copyright 2025 by the author(s).

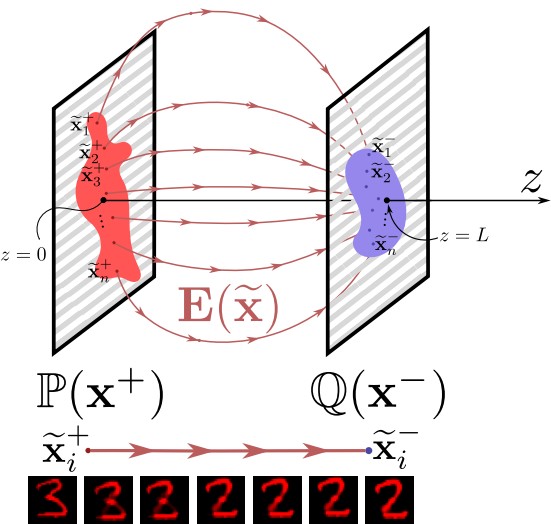

*Figure 1.* Our Electrostatic field matching (EFM) method. Two data distributions $\mathbb{P}(\mathbf{x}^+)$ and $\mathbb{Q}(\mathbf{x}^-)$, $\mathbf{x}^\pm \in \mathbb{R}^D$ are placed in the space $\mathbb{R}^{D+1}$ in the planes $z = 0$ and $z = L$, respectively. The distribution $\mathbb{P}(\mathbf{x}^+)$ is assigned a positive charge, and the distribution $\mathbb{Q}(\mathbf{x}^-)$ – a negative charge. These charges create an electric field $\mathbf{E}(\widetilde{\mathbf{x}})$, where $\widetilde{\mathbf{x}} = (\mathbf{x}, z) \in \mathbb{R}^{D+1}$. The lines of the field begin at positive charges and end at negative charges. Movement along the electric field lines provably (see our Theorem 3.1) transforms the distribution $\mathbb{P}(\mathbf{x}^+)$ into the distribution $\mathbb{Q}(\mathbf{x}^-)$.

**Diffusion Models** (Sohl-Dickstein et al., 2015; Ho et al., 2020, DM) is a popular class of generative models which is inspired by the *nonequilibrium thermodynamics*. The diffusion models consist of forward and backward stochastic processes (Song et al., 2021). The forward process corrupts the data via injection of Gaussian noise; the backward process reverses the forward process and recovers the data.

**Poisson Flow Generative Models** (Xu et al., 2022; 2023, PFGM) use ideas from the *electrostatic* theory for the data generation process, recovering an electric field between a hyperplane of the data and a hemisphere of large radius.

Both DM and PFGM use physical principles to corrupt data, simplifying the data distribution to a tractable one. As a result, they are only used directly for **noise-to-data** tasks.

Recently, modifications of DM have appeared that can learn diffusion in a data-to-data scenario. Diffusion Bridge Match-

ing (Shi et al., 2024; Albergo & Vanden-Eijnden, 2023; Peluchetti, 2023, BM) is an SDE-based method that recovers the continuous-time Markovian process between data distributions. Flow matching (Lipman et al., 2023; Liu et al., 2023; Klein et al., 2024; Chen & Lipman, 2024; Xie et al., 2024, FM) is the limiting case of BM that learns ODE-based transformation between distributions.

However, there is no method based on electrostatic theory that can be applied to **data-to-data** translation tasks.

**Contributions.** We propose and theoretically justify a new paradigm for generative modeling called *Electrostatic Field Matching* (EFM). It is based on the electrostatic theory and suitable for both noise-to-data and data-to-data generative scenarios. We provide proof-of-concept experiments on low- and high-dimensional generative modeling tasks.

## 2. Background and Related Works

### 2.1. Basic physics

To understand the physics behind the electrostatic field matching method, let us recall some basic background from standard Maxwell's 3D-electrostatics and then generalize it to the case of $D$ dimensions. Information on Maxwell's electrostatics can be found in any electricity textbook, for instance (Landau & Lifshitz, 1971, Chapter 5).

#### 2.1.1. MAXWELL'S ELECTROSTATICS[1]

**The field of a point charge.** Let a point charge $q \in \mathbb{R}$ be located at a point $\mathbf{x}' \in \mathbb{R}^3$. At a point $\mathbf{x} \in \mathbb{R}^3$ it creates an electric field[2] $\mathbf{E}(\mathbf{x}) \in \mathbb{R}^3$ equal to:

$$\mathbf{E}(\mathbf{x}) = \frac{q}{4\pi} \frac{\mathbf{x} - \mathbf{x}'}{||\mathbf{x} - \mathbf{x}'||^3}. \tag{1}$$

**The superposition principle.** If point charges $q_1, q_2, ..., q_N$ are located at points $\mathbf{x}_1, \mathbf{x}_2, ..., \mathbf{x}_N$, they create independent fields $\mathbf{E}_1(\mathbf{x}), \mathbf{E}_2(\mathbf{x}), ..., \mathbf{E}_N(\mathbf{x})$ at a given point $\mathbf{x} \in \mathbb{R}^3$. All these charges together produce the following field:

$$\mathbf{E}(\mathbf{x}) = \sum_{n=1}^{N} \mathbf{E}_n(\mathbf{x}) = \sum_{n=1}^{N} \frac{q_n}{4\pi} \frac{(\mathbf{x} - \mathbf{x}_n)}{||\mathbf{x} - \mathbf{x}_n||^3}, \tag{2}$$

In the general case, we are dealing with a continuously distributed charge $q(\mathbf{x})$. Then the superposition principle

---

[1]All formulas are written in the Heaviside–Lorentz system of units, where Planck's constant $\hbar = 1$, the speed of light $c = 1$, and the electric constant, which stands as a multiplier in Coulomb's law (see (1)), is $k = 1/(4\pi)$. This system of units is convenient for our purposes because formulas look particularly simple in this system of units (the Gauss's theorem and the circulation theorem).

[2]The meaning of electric field is as follows. If a charge $q_0$ is placed in an electric field, then the force acting on $q_0$ equals to $\mathbf{F} = q_0 \mathbf{E}$. Using (1), we obtain Coulomb's law of interaction of point charges: $\mathbf{F} = k \frac{q q_0}{||\mathbf{x} - \mathbf{x}'||^3}(\mathbf{x} - \mathbf{x}')$, where $k = \frac{1}{4\pi}$.

can be written as:

$$\mathbf{E}(\mathbf{x}) = \int \frac{1}{4\pi} \frac{(\mathbf{x} - \mathbf{x}')}{||\mathbf{x} - \mathbf{x}'||^3} q(\mathbf{x}') d\mathbf{x}'. \tag{3}$$

The charge distribution $q(\mathbf{x})$ can have values greater than zero (positive charge) or less than zero (negative charge).

**An electric field strength line** is a curve $\mathbf{x}(t) \in \mathbb{R}^3$, $t \in [a, b] \subset \mathbb{R}$ whose tangent to each point is parallel to the electric field at that point. In other words:

$$\frac{d\mathbf{x}(t)}{dt} = \mathbf{E}(\mathbf{x}(t)), \text{where } t \in [a, b] \subset \mathbb{R}. \tag{4}$$

Electric field lines are the key concept in our work. The basic properties of these lines follow from Gauss's theorem and the circulation theorem formulated below. The rigorous formulations and proofs of the field line properties are given in Appendix A (for a more general $D$-dimentional case).

**The electric field flux.** Consider an element of area $\mathbf{dS}$. It is a *vector* whose length is equal to the considered area $dS$ and whose direction is orthogonal to this area. For closed surfaces, the direction is always selected outward. The electric field flux $\mathbf{E}$ through this element is $d\Phi = \mathbf{E} \cdot \mathbf{dS}$, i.e., the inner product between vectors $\mathbf{E}$ and $\mathbf{dS}$. The field flux through a finite surface $\Sigma$ is given by

$$\Phi = \int_{\Sigma} d\Phi = \iint_{\Sigma} \mathbf{E} \cdot \mathbf{dS}. \tag{5}$$

It indicates the density of field lines passing through $\Sigma$.

**Gauss's theorem** (Landau & Lifshitz, 1971, §31). For any closed two-dimensional surface $\partial M$, which bounds the set $M \subset \mathbb{R}^3$ (see Fig. 2), the electric field flux is equal to the total charge enclosed by this surface:

$$\iint_{\partial M} \mathbf{E} \cdot \mathbf{dS} = \int_{M} q(\mathbf{x}) d\mathbf{x}. \tag{6}$$

Intuitively, the Gauss's theorem states that the number of lines passing through any closed surface is determined only by the charge inside that surface (and is proportional to it).

In particular, it follows from Gauss's theorem that the electric field line must begin at a positive charge (or at infinity) and end at a negative charge (or at infinity). *Lines cannot simply terminate in a space where there are no charges*, see Lemma A.3 and Corollary A.4 for extended discussions.

**Theorem on the electric field circulation** (Landau & Lifshitz, 1971, §26). For any closed loop $\ell$ (Fig. 3) the electric field circulation is equal to zero:

$$\oint_{\ell} \mathbf{E} \cdot \mathbf{dl} = 0, \tag{7}$$

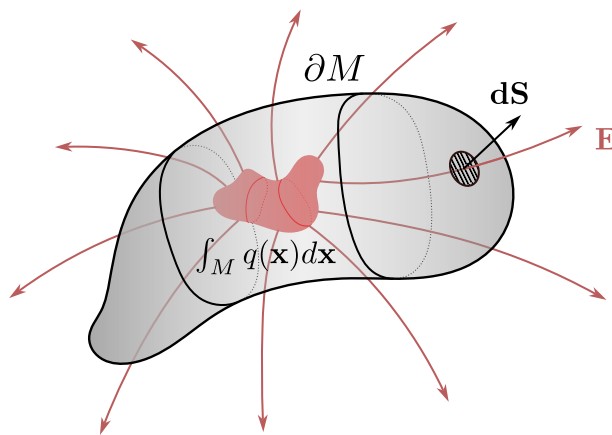

*Figure 2.* An illustration of the Gauss's theorem.

where **dl** is the length element of the closed loop $\ell$. This length element is a *vector* whose length is determined by the infinitesimal segment **dl**, see Fig. 3, whose direction is the tangent at a given point to the curve $\ell$.

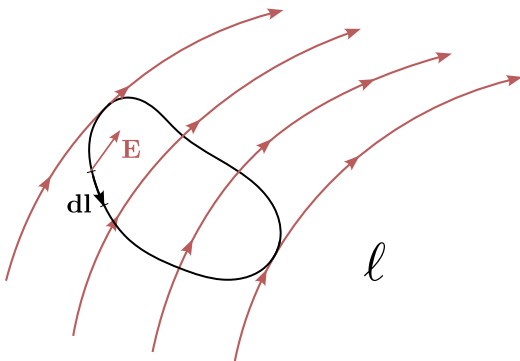

*Figure 3.* An illustration of the electric field circulation theorem.

It follows from the circulation theorem that *there are no field lines which form closed loops*, see Lemma A.7.

### 2.1.2. $D$-DIMENSIONAL ELECTROSTATICS

The generalization of electrostatic equations for higher dimensions appears in discussions related to the influence of extra dimensions on physics (Ehrenfest, 1917; Gurevich & Mostepanenko, 1971; Caruso et al., 2023). The generalization modifies equations (6) and (7) by replacing $\mathbb{R}^3$ with $\mathbb{R}^D$ and replacing dimensionality 2 of $\partial M$ with $D-1$ in the Gauss's theorem. The definitions in (4) and the superposition principle remain unchanged. The differences affect only the explicit expression for the electric field.

The electric field at the point $\mathbf{x} \in \mathbb{R}^D$ of a point charge $q$, which is located at $\mathbf{x}' \in \mathbb{R}^D$ equals to:

$$\mathbf{E}(\mathbf{x}) = \frac{q}{S_{D-1}} \frac{\mathbf{x} - \mathbf{x}'}{||\mathbf{x} - \mathbf{x}'||^D}, \qquad (8)$$

where $S_{D-1}$ is the surface area of an $(D-1)$-dimensional sphere with radius 1.

The field of a distributed charge $q(\mathbf{x})$ can be obtained by the principle of superposition as in (3) for 3D case:

$$\mathbf{E}(\mathbf{x}) = \int \frac{1}{S_{D-1}} \frac{\mathbf{x} - \mathbf{x}'}{||\mathbf{x} - \mathbf{x}'||^D} q(\mathbf{x}')d\mathbf{x}'. \qquad (9)$$

The Gauss's theorem and the circulation theorem in a $D$-dimensional space together ensure the following **principal characteristics** of electric field lines:

**(i)** Electric field lines cannot terminate in points where there are no charges;

**(ii)** For a system having zero total charge ($\int q(\mathbf{x})d\mathbf{x} = 0$), electric field lines almost surely[3] start at positive charge and end at negative charge;

**(iii)** There are no electric field lines that form closed loops.

### 2.2. Poisson Flow Generative Model (PFGM)

The first attempt to couple electrostatic theory and generative modeling is proposed by (Xu et al., 2022; 2023). The authors work with a $D$-dimensional data distribution.

They embed this distribution into $(D+1)$-dimensional space. The new point in this space can be written as

$$(x_1, x_2, ..., x_D, z) = (\mathbf{x}, z) = \widetilde{\mathbf{x}} \in \mathbb{R}^{D+1}. \qquad (10)$$

The data is then placed on the hyperplane $z = 0$ at $\mathbb{R}^{D+1}$ by applying $\mathbf{x} \to \widetilde{\mathbf{x}} = (\mathbf{x}, 0)$. The data distribution is interpreted then as a positive electrostatic charge distribution.

The intuition of the method is that the charged points $\widetilde{\mathbf{x}}$ in the hyperplane $z = 0$ generate the electric field $\mathbf{E}(\cdot)$ which behaves at infinity as the field of a point charge. If a point charge is placed inside a sphere $\mathcal{S}_\infty$ with an infinite radius, then the flux density $\mathbb{P}_\infty(\cdot)$ through the surface of the sphere is distributed uniformly. For simplicity, the authors consider the hemisphere $\mathcal{S}_\infty^+$ (Fig. 4b, upper half of $\mathcal{S}_\infty$). The electric field lines define the correspondence between uniformly distributed charges on the surface of $\mathcal{S}_\infty^+$ and the data distribution $\mathbb{P}_0(\widetilde{\mathbf{x}})$ located in the hyperplane $z = 0$.

If a massless point charge is placed in the electric field $\mathbf{E}(\cdot)$ with field lines directed from $\mathbb{P}_0(\cdot)$ to $\mathbb{P}_\infty(\cdot)$, then the charge moves along the lines to $\mathbb{P}_\infty(\cdot)$. This movement transforms the data samples from the complex distribution $\mathbb{P}_0(\cdot)$ into

---

[3]The term "almost surely" indicates that for a randomly selected point from the first positively charged distribution, the probability that its electric field line terminates at the second negatively charged distribution (rather than at infinity) equals 1.

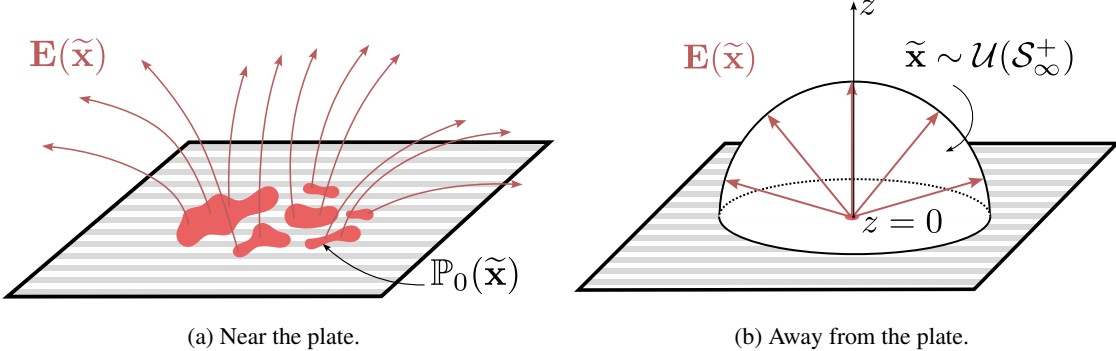

(a) Near the plate.

(b) Away from the plate.

*Figure 4.* PFGM concept. The original data have a distribution $\mathbb{P}_0(\widetilde{\mathbf{x}})$, which is assigned a positive charge that produces an electric field $\mathbf{E}(\widetilde{\mathbf{x}})$. Near the plate (Fig. 4a), the field lines can have a complex structure, while away from the plate (Fig. 4b) the charge looks like a point, and therefore the electric field is uniformly distributed: $\widetilde{\mathbf{x}} \sim \mathcal{U}(\mathcal{S}_\infty^+)$.

the simple distribution $\mathbb{P}_\infty(\cdot)$ on the hemisphere. The corresponding inverse transformation generates the data samples from uniformly distributed samples on the hemisphere. The inverse map is a movement along these field lines in the backward direction and is defined by the following ODE with electric field $-\mathbf{E}(\cdot)$ being the velocity field:

$$\frac{d\widetilde{\mathbf{x}}(t)}{dt} = -\mathbf{E}(\widetilde{\mathbf{x}}(t)). \tag{11}$$

To recover the electric field $\mathbf{E}(\cdot)$ in the extended $(D+1)$-dimensional space, the authors propose to approximate it with a neural network $f_\theta(\cdot) : \mathbb{R}^{D+1} \to \mathbb{R}^{D+1}$.

First, they compute the ground truth electric field $\mathbf{E}(\widetilde{\mathbf{x}})$ empirically at a set of arbitrary $(D+1)$-dimensional points $\widetilde{\mathbf{x}}$ inside the hemisphere $\mathcal{S}_\infty^+$ through samples from $\mathbb{P}_0(\cdot)$ using (9). Second, the electric field is learned at $\widetilde{\mathbf{x}}$ by minimizing the difference between the predicted $f_\theta(\widetilde{\mathbf{x}})$ and the ground-truth $\mathbf{E}(\widetilde{\mathbf{x}})$. Having learned the electric field $\mathbf{E}(\cdot)$ in the $(D+1)$-dimensional space, they simulate ODE (11) with initial samples from $\mathbb{P}_\infty(\cdot)$ until the spatial coordinate $z$ reaches 0. Finally, they get samples $\widetilde{\mathbf{x}}_T \sim \mathbb{P}_0(\cdot)$, where $T$ is the end time of the ODE simulation.

# 3. Electrostatic Field Matching (EFM)

This section introduces Electrostatic Field Matching (EFM), a novel generative modeling paradigm applicable to **both** noise-to-data and data-to-data generation grounded in electrostatic theory. In §3.1, we give an intuitive description of the method. In §3.2, we give the theoretical foundation of the method and the main theorem. In §3.3, we formulate the learning and inference algorithms of the EFM.

### 3.1. Intuitive explanation of the method

Our idea is to consider distributions as electric charge densities. One could assign positive charge values to the first distribution and negative charges to the second one, i.e., the

charge density follows the distributions up to a sign. We then place these distributions on two $D$-dimensional planes at distance $L$ from each other (Fig. 1). This will produce an electric field with lines starting at one density and finishing at another. We prove Theorem 3.1 — our key result — which shows that movement along the lines guarantees an almost sure transition from one distribution to another.

### 3.2. Formal theoretical justification

Let $\mathbb{P}(\mathbf{x}^+)$ and $\mathbb{Q}(\mathbf{x}^-)$, with $\mathbf{x}^\pm \in \mathbb{R}^D$, be two data distributions. We assign to the first distribution a positive charge $q^+(\mathbf{x}^+) = \mathbb{P}(\mathbf{x}^+)$ and to the second distribution a negative charge $q^-(\mathbf{x}^-) = -\mathbb{Q}(\mathbf{x}^-)$. Note that the charge distributions are normalized such that $\int q^+(\mathbf{x}^+)d\mathbf{x}^+ = 1$ and $\int q^-(\mathbf{x}^-)d\mathbf{x}^- = -1$, resulting in a total charge of zero.

We place these $q^+$ and $q^-$ in $(D+1)$-dimensional space. Each point in this space has the same form as in (10). More precisely, we place $q^+$ in the hyperplane $z = 0$, and $q^-$ in $z = L$ (Fig. 1). One can think of it as a $(D+1)$-dimensional **capacitor**. The distributions can be written with Dirac delta function $\delta(\cdot)$ as:

$$\begin{aligned} q^+(\widetilde{\mathbf{x}}) = q^+(\mathbf{x}, z) = q^+(\mathbf{x})\delta(z), \\ q^-(\widetilde{\mathbf{x}}) = q^-(\mathbf{x}, z) = q^-(\mathbf{x})\delta(z - L). \end{aligned} \tag{12}$$

The electric field produced at the point $\widetilde{\mathbf{x}} \in \mathbb{R}^{D+1}$ in the space plates consists of two summands:

$$\mathbf{E}(\widetilde{\mathbf{x}}) = \mathbf{E}_+(\widetilde{\mathbf{x}}) + \mathbf{E}_-(\widetilde{\mathbf{x}}), \tag{13}$$

where $\mathbf{E}_+(\mathbf{x})$ and $\mathbf{E}_-(\widetilde{\mathbf{x}})$ are the fields created by $q^+(\widetilde{\mathbf{x}}^+)$ and $q^-(\widetilde{\mathbf{x}}^-)$, respectively. The exact expression for these fields is given by (9) with replacement of $D$ by $D+1$:

$$\mathbf{E}_\pm(\widetilde{\mathbf{x}}) = \int \frac{1}{S_D} \frac{\widetilde{\mathbf{x}} - \widetilde{\mathbf{x}}'}{||\widetilde{\mathbf{x}} - \widetilde{\mathbf{x}}'||^{D+1}} q^\pm(\widetilde{\mathbf{x}}')d\widetilde{\mathbf{x}}'. \tag{14}$$

Consider the field lines originating from the plate $\mathbb{P}(\widetilde{\mathbf{x}}^+)$. Due to the total zero charge, these lines almost surely terminate on $\mathbb{Q}(\widetilde{\mathbf{x}}^-)$, as established in Lemma A.6. The transport from $\mathbb{P}(\cdot)$ to $\mathbb{Q}(\cdot)$ can be initiated in two distinct directions: either toward (forward-oriented) or away from (backward-oriented) the target plate. While backward-oriented lines exhibit greater curvature and longer paths, they still terminate in the target distribution, see Figs. 21a and 21b.

We define the *stochastic* **forward map** $T_F$ from $\mathrm{supp}(\mathbb{P})$ to $\mathrm{supp}(\mathbb{Q})$ through forward-oriented field lines.[4] For this, we consider a point $\widetilde{\mathbf{x}}^+ = (\mathbf{x}^+, \varepsilon), \varepsilon \to 0^+$ slightly shifted in the direction of the second plate. Let us move along the corresponding field line by integrating $d\widetilde{\mathbf{x}}(t) = \mathbf{E}(\widetilde{\mathbf{x}}(t))dt$ until coming to the second plate at a point $\widetilde{\mathbf{x}}_F^- = (\mathbf{x}_F^-, L)$. Next, two situations may arise (Fig. 7):

1. The values of the field to the left and right of the plate $z = L$ have different signs. Here the field is directed toward the plate from two different sides.

2. The values of the field to the left and right of the plate have the same signs on the left and right, i.e., the field line crosses $z = L$ and continues to the region $z > L$.[5]

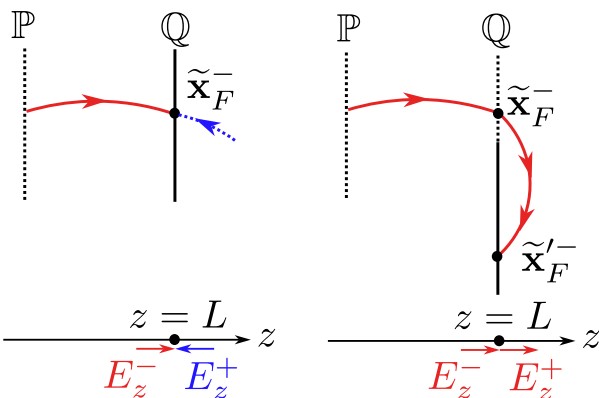

*Figure 5.* An illustration of the forward map $T_F$.

In the first case, the movement ends. In the second case, it is necessary to stop the movement at the point $\widetilde{\mathbf{x}}_F^-$ with probability $\nu(\mathbf{x}_F^-)$ and continue the movement into the region $z > L$ with probability $1 - \nu(\mathbf{x}_F^-)$, where

$$\nu(\mathbf{x}_F^-) = \begin{cases} 1, & \text{if } E_z^\pm(\mathbf{x}_F^-) \text{ have opposite signs} \\ \frac{E_z^-(\mathbf{x}_F^-) - E_z^+(\mathbf{x}_F^-)}{E_z^-(\mathbf{x}_F^-)} & \text{if } E_z^\pm(\mathbf{x}_F^-) \text{ have same sign} \end{cases} \tag{15}$$

and $E_z^\pm(\mathbf{x}_F^-) = E_z(\widetilde{\mathbf{x}}_F^- \pm \varepsilon \mathbf{e}_z), \varepsilon \to 0^+$ denote the left and right limits of the field value at the point $\widetilde{\mathbf{x}}_F^-$ in $z$ direction. The latter field line also reaches the target distribution $\mathbb{Q}$, at

a point $\widetilde{\mathbf{x}}_F^{\prime -}$. Finally, we define the forward map as follows:

$$T_F(\mathbf{x}^+) = \begin{cases} \mathbf{x}_F^- & \text{with probability } \nu(\mathbf{x}_F^-) \\ \mathbf{x}_F^{\prime -} & \text{with probability } 1 - \nu(\mathbf{x}_F^-) \end{cases}. \tag{16}$$

The probability $\nu(\mathbf{x}_F^-)$ allows us to understand whether it is needed to stop at the first intersection point $\mathbf{x}_F^-$, or continue moving to the next point $\mathbf{x}_F^{\prime -}$ (see Fig. 5). In particular, the probability value is proportional to the electric field flux.

The *stochastic* **backward map** $T_B$ is constructed similarly using left limit $\varepsilon \to 0^-$ and backward-oriented field lines.

The complete transport is described by the *random variable*:

$$T(\mathbf{x}^+) = \begin{cases} T_F(\mathbf{x}^+) \text{ with probability } \mu(\mathbf{x}^+), \\ T_B(\mathbf{x}^+) \text{ with probability } 1 - \mu(\mathbf{x}^+), \end{cases} \tag{17}$$

capturing both forward and backward trajectory endpoints for each $\mathbf{x}^+ \in \mathrm{supp}(\mathbb{P})$ on the left plate, where:

$$\mu(\mathbf{x}^+) = \begin{cases} 1, & \text{if } E_z^\pm(\mathbf{x}^+) \text{ have same sign} \\ \frac{E_z^+(\mathbf{x}^+)}{E_z^+(\mathbf{x}^+) + |E_z^-(\mathbf{x}^+)|} & \text{if } E_z^\pm(\mathbf{x}^+) \text{ have opp. signs} \end{cases} \tag{18}$$

The value $\mu(\mathbf{x}^+)$ allows one to choose a forward or backward sets of lines with a probability proportional to the field flux in the corresponding direction (i.e., $E_z^+$ or $|E_z^-|$). At the same time, if it is impossible to move backward, i.e., $E_z^- > 0$, the forward map $T_F$ is always chosen ($\mu(\mathbf{x}^+) = 1$).

Then, for this stochastic map, we prove the next theorem:

**Theorem 3.1** (**Electrostatic Field Matching**). *Let $\mathbb{P}(\boldsymbol{x}^+)$ and $\mathbb{Q}(\boldsymbol{x}^-)$ be two data distributions that have compact support. Let $\boldsymbol{x}^+$ be distributed as $\mathbb{P}(\boldsymbol{x}^+)$. Then $\boldsymbol{x}^- = T(\boldsymbol{x}^+)$ is distributed as $\mathbb{Q}(\boldsymbol{x}^-)$ almost surely:*

$$\text{If } \boldsymbol{x}^+ \sim \mathbb{P}(\boldsymbol{x}^+) \Rightarrow T(\boldsymbol{x}^+) = \boldsymbol{x}^- \sim \mathbb{Q}(\boldsymbol{x}^-). \tag{19}$$

In other words, the movement along electrostatic field lines does indeed transfer $\mathbb{P}(\mathbf{x}^+)$ to $\mathbb{Q}(\mathbf{x}^-)$, as we intended. The proof of the theorem is given in Appendix B.

### 3.3. Learning and Inference Algorithm

To move between data distributions, it is sufficient to follow the electric field lines. The lines can be found from the trained neural network approximating the electric field $\mathbf{E}(\widetilde{\mathbf{x}})$. In practice, we only use forward-oriented field lines, since backward-oriented ones have more curvature and require a larger training volume, see Appendices C.1 and C.3.

**Training.** To recover the electric field $\mathbf{E}(\cdot)$ in $(D+1)$-dimensional points between the hyperplanes, we approximate it with a neural network $f_\theta(\cdot) : \mathbb{R}^{D+1} \to \mathbb{R}^{D+1}$. The approximation requires setting of a training volume. We sample the value $t$ from the uniform distribution $\mathcal{U}(0, L)$

---

[4]Here $\mathrm{supp}(\cdot)$ denotes the support of a distribution.

[5]The movement in the opposite direction is impossible since the projection $E_z$ of the field is positive in the interval $z \in (0, L)$.

and take two random samples $\widetilde{\mathbf{x}}^+$ and $\widetilde{\mathbf{x}}^-$. Then, we get a new point $\widetilde{\mathbf{x}}$ via the following averaging and noising:

$$\widetilde{\mathbf{x}} = \frac{t}{L}\widetilde{\mathbf{x}}^- + (1 - \frac{t}{L})\widetilde{\mathbf{x}}^+ + \widetilde{\varepsilon}, \qquad (20)$$

where random $\widetilde{\varepsilon}$ is obtained as follows. First, the noise $\varepsilon$ is sampled from $\mathcal{N}(\frac{L}{2}, \sigma^2 I_{(D+1)\times(D+1)})$, where $\sigma$ is a hyperparameter. Then, the Euclidean norm $||\varepsilon||$ is multiplied by a normalized standard Gaussian vector $m$:

$$\widetilde{\varepsilon} = ||\varepsilon||\frac{m}{||m||}, \quad m \sim \mathcal{N}(0, I_{(D+1)\times(D+1)}). \qquad (21)$$

The noise term increases the training volume, thus, leading to the greater generalization ability of the neural network. We highlight that this interpolation is just one of the possible ways to define intermediate points, see Appendix C.4.

The ground-truth $\mathbf{E}(\widetilde{\mathbf{x}})$ is estimated with (13). Specifically, the integral is approximated via Monte Carlo sampling of (9) by using samples from $\mathbb{P}(\mathbf{x}^+)$ and $\mathbb{Q}(\mathbf{x}^-)$. Then we use a neural network approximation $f_\theta(\widetilde{\mathbf{x}})$ to learn the *normalized* ground-truth electric field $\frac{\mathbf{E}(\widetilde{\mathbf{x}})}{||\mathbf{E}(\widetilde{\mathbf{x}})||}$ at points from the extended $(D+1)$-dimensional space as, analogously to PFGM, we empirically found that this strategy works better than learning the unnormalized field. We learn $f_\theta(\cdot)$ by minimizing the squared error between the normalized ground truth $\frac{\mathbf{E}(\widetilde{\mathbf{x}})}{||\mathbf{E}(\widetilde{\mathbf{x}})||}$ and the predictions $f_\theta(\widetilde{\mathbf{x}})$ with SGD, i.e., the learning objective is

$$\mathbb{E}_{\widetilde{\mathbf{x}}}||f_\theta(\widetilde{\mathbf{x}}) - \frac{\mathbf{E}(\widetilde{\mathbf{x}})}{||\mathbf{E}(\widetilde{\mathbf{x}})||}||^2 \to \min_\theta. \qquad (22)$$

**Inference.** Having learned the normalized vector field $\frac{\mathbf{E}(\cdot)}{||\mathbf{E}(\cdot)||}$ in the extended space with $f_\theta(\cdot)$, we simulate the movement between hyperplanes to transfer data from $\mathbb{P}(\mathbf{x}^+)$ to $\mathbb{Q}(\mathbf{x}^-)$. For this, we run an ODE solver for (11).

One needs a right stopping time for the ODE solver. We follow the idea of PFGM and exploit the following formula:

$$d\widetilde{\mathbf{x}} = d(\mathbf{x}, z) = \left(\frac{d\mathbf{x}}{dt}\frac{dt}{dz}dz, dz\right) = (\mathbf{E}_x(\widetilde{\mathbf{x}})\mathbf{E}_z^{-1}(\widetilde{\mathbf{x}}), 1)dz$$
$$= \left(\frac{\mathbf{E}_x(\widetilde{\mathbf{x}})}{||\mathbf{E}(\widetilde{\mathbf{x}})||}\frac{||\mathbf{E}(\widetilde{\mathbf{x}})||}{\mathbf{E}_z(\widetilde{\mathbf{x}})}, 1\right)dz \approx (f_\theta(\widetilde{\mathbf{x}})_x f_\theta^{-1}(\widetilde{\mathbf{x}})_z, 1)dz \qquad (23)$$

where we denote $f_\theta(\widetilde{\mathbf{x}}) = (f_\theta(\widetilde{\mathbf{x}})_x, f_\theta(\widetilde{\mathbf{x}})_z)$ and $\mathbf{E}(\widetilde{\mathbf{x}}) = (\mathbf{E}_x(\widetilde{\mathbf{x}}), \mathbf{E}_z(\widetilde{\mathbf{x}}))$. In this new ODE (23), we replace the time variable $t$ with the physically meaningful variable $z$. We start with samples from $\mathbb{P}(\mathbf{x}^+)$, i.e., when $z = 0$, and we stop when $z$ reaches $L$ (= right plate) during the ODE path.

According to definition (16) of the forward map $T_F$, when field line crosses the plane $z = L$, one should continue its movement with probability $1 - \nu(\mathbf{x}_F^-)$ until the line

reaches $z = L$ again. However, in practice we simply stop the movement as accurate estimation of this probability is non-trivial, see Appendix C.2 for extended discussions.

All the ingredients for training and inference in our method are described in Algorithms 1 and 2, where we summarize the learning and the inference procedures, correspondingly.

---

**Algorithm 1** EFM Training

**Input:** Distributions accessible by samples:
 $\mathbb{P}(\mathbf{x}^+)\delta(z)$ and $\mathbb{Q}(\mathbf{x}^-)\delta(z - L)$;
 NN approximator $f_\theta(\cdot) : \mathbb{R}^{D+1} \to \mathbb{R}^{D+1}$;
**Output:** The learned normalized electrostatic field $f_\theta(\cdot)$
**Repeat until converged :**
 Sample a batch of points $\widetilde{\mathbf{x}}^+ \sim \mathbb{P}(\mathbf{x}^+)\delta(z)$;
 Sample a batch of points $\widetilde{\mathbf{x}}^- \sim \mathbb{Q}(\mathbf{x}^-)\delta(z - L)$;
 Sample a batch of times $t \sim \mathcal{U}(0, L)$;
 Sample a batch of noise $\widetilde{\varepsilon}$ with (21);
 Calculate $\widetilde{\mathbf{x}} = \frac{t}{L}\widetilde{\mathbf{x}}^+ + (1 - \frac{t}{L})\widetilde{\mathbf{x}}^- + \widetilde{\varepsilon}$;
 Estimate $\mathbf{E}_+(\widetilde{\mathbf{x}})$ and $\mathbf{E}_-(\widetilde{\mathbf{x}})$ through (9);
 Calculate $\mathbf{E}(\widetilde{\mathbf{x}})$ with (13);
 Compute $\mathcal{L} = \mathbb{E}_{\widetilde{\mathbf{x}}}||f_\theta(\widetilde{\mathbf{x}}) - \frac{\mathbf{E}(\widetilde{\mathbf{x}})}{||\mathbf{E}(\widetilde{\mathbf{x}})||}||^2 \to \min_\theta$;
 Update $\theta$ by using $\frac{\partial \mathcal{L}}{\partial \theta}$;

---

**Algorithm 2** EFM Sampling

**Input:** sample $\widetilde{\mathbf{x}}^+$ from $\mathbb{P}(\mathbf{x}^+)\delta(z)$; step size $\Delta\tau > 0$;
  the learned field $f_\theta^*(\cdot) : \mathbb{R}^{D+1} \to \mathbb{R}^{D+1}$;
**Output:** mapped sample $\widetilde{\mathbf{x}}^-$ approximating $\mathbb{Q}(\mathbf{x}^-)\delta(z - L)$
Set $\widetilde{\mathbf{x}}_0 \leftarrow \widetilde{\mathbf{x}}^+$
**for** $\tau \in \{0, \Delta\tau, 2\Delta\tau, \dots, L - \Delta\tau\}$ **do**
 Calculate $f_\theta^*(\widetilde{\mathbf{x}}_\tau) = (f_\theta^*(\widetilde{\mathbf{x}}_\tau)_x, f_\theta^*(\widetilde{\mathbf{x}}_\tau)_z)$
 $\widetilde{\mathbf{x}}_{\tau+\Delta\tau} \leftarrow \left[(\widetilde{\mathbf{x}}_\tau)_x + f_\theta^*(\widetilde{\mathbf{x}}_\tau)_z^{-1} f_\theta^*(\widetilde{\mathbf{x}}_\tau)_x \Delta\tau; \tau + \Delta\tau\right]$
$\widetilde{\mathbf{x}}^- \leftarrow \widetilde{\mathbf{x}}_L$

---

## 4. Experimental Illustrations

In this section, we demonstrate the proof-of-concept experiments with our proposed EFM method. We show a 2-dimensional illustrative experiment (§4.1), image-to-image translation experiment (§4.2) and image generation experiment (§4.3) with the colored MNIST and CIFAR-10 datasets. Various additional aspects of EFM are studied in Appendices: the influence of the interplate distance $L$ (App. C.1), training volume (App. C.4) and backward-oriented field lines (App. C.3). We give details of experiments in Appendix D.

### 4.1. Gaussian to Swiss Roll Experiment

An intuitive first test to validate the method is to transfer between distributions whose densities can be visualized for comparison. We consider the 2-dimensional zero-centered Gaussian distribution with the identity covariance matrix as

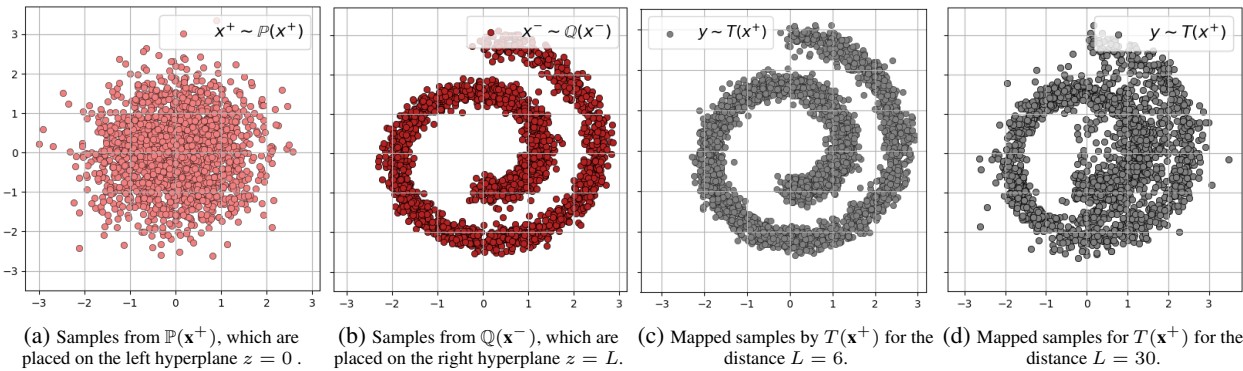

(a) Samples from $\mathbb{P}(\mathbf{x}^+)$, which are placed on the left hyperplane $z = 0$.

(b) Samples from $\mathbb{Q}(\mathbf{x}^-)$, which are placed on the right hyperplane $z = L$.

(c) Mapped samples by $T(\mathbf{x}^+)$ for the distance $L = 6$.

(d) Mapped samples for $T(\mathbf{x}^+)$ for the distance $L = 30$.

*Figure 6. Illustrative 2D Gaussian→Swiss Roll experiment*: input and target distributions $\mathbb{P}(\mathbf{x}^+)$ and $\mathbb{Q}(\mathbf{x}^-)$ together with the result of the distribution transfer learned with our EFM method for distances $L = 6$ and $L = 30$ between the capacitor plates.

$\mathbb{P}(\mathbf{x}^+)$ and the Swiss Roll distribution as $\mathbb{Q}(\mathbf{x}^-)$, see their visualizations in Figs. 6a and 6b, respectively.

To show the effect of hyperparameter $L$ in our EFM method, we do two experiments. In the first one, the samples from $\mathbb{Q}(\mathbf{x}^-)$ are placed on the hyperplane $L = 6$ (see Fig. 6c), while in the second one, we use $L = 30$ (see Fig. 6d). We show the learned trajectories of samples' movement along the electrostatic field in Figs. 8a and 8b, respectively.

When $L$ is small, the electric field lines are more or less straight, see Fig. 8a. The learned normalized electric field $f_\theta(\cdot)$ allows one to accurately perform the distribution transfer, see Fig. 6c. However, if the distance $L$ between the hyperplanes is large, the learned map recovers the target density $\mathbb{Q}(\mathbf{x}^-)$ poorly (see Fig. 6d). Presumably, this occurs due to a higher difficulty of the electrical field $\mathbf{E}(\cdot)$ approximation for a large interplane distance $L$, as well as the neccesity to consider the lines outside $z \in (0, L)$, i.e., backward-oriented lines as well as the forward-oriented lines which go to the area $z > L$ (intersect the plate twice).

### 4.2. Image-to-Image Translation Experiment

Here we consider the image-to-image translation task for transforming colored digits 3 to colored digits 2 (Gushchin et al., 2024, §5.3). The data is based on the conventional MNIST images dataset but the digits are randomly colored. We consider *unpaired* translation task, i.e., there is no pre-

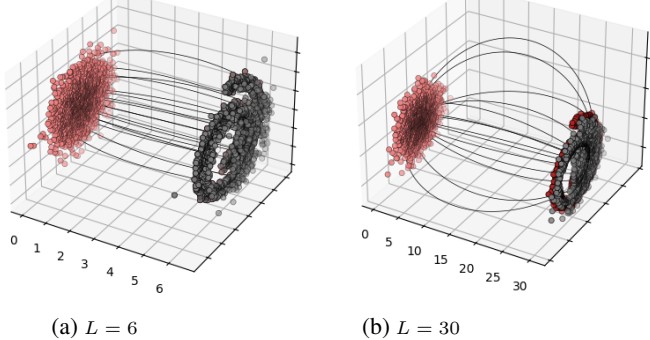

(a) $L = 6$

(b) $L = 30$

*Figure 8.* Electric field line structure for the Gaussian→Swiss Roll experiment with $L = 6$ and $L = 30$. It can be seen that at large distances, the field lines are more curved than at small distances.

defined correspondence, see (Zhu et al., 2017, Fig. 2).

We place colored digits 3 on the left hyperplane $z = 0$ and colored digits 2 on the right plate $z = 10$. We learn the normalized electric field between the plates and demonstrate input-translated pairs in Fig. 10a and 10b, respectively. Also, we show how the translation happens in Fig. 7.

For comparison, we add the results of the translation of the popular ODE-based Flow Matching (FM) method (Liu et al., 2023; Lipman et al., 2023; Tong et al., 2023) in Fig. 10c. The key difference between our method and FM is that FM matches to a *time*-conditional transformation (velocity), whereas our method matches to a *space*-conditional transformation (electric field). Interestingly, FM does not always

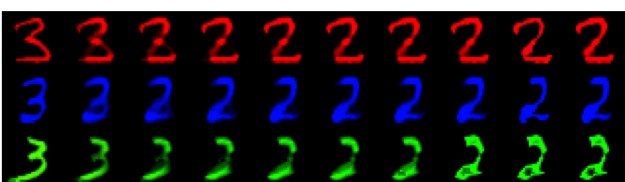

*Figure 7.* The sampling trajectories of our **EFM** method in image-to-image translation experiment, see §4.2.

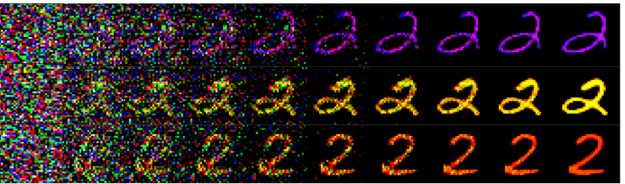

*Figure 9.* The sampling trajectories of our **EFM** method in noise-to-image generation experiment, see §4.3.

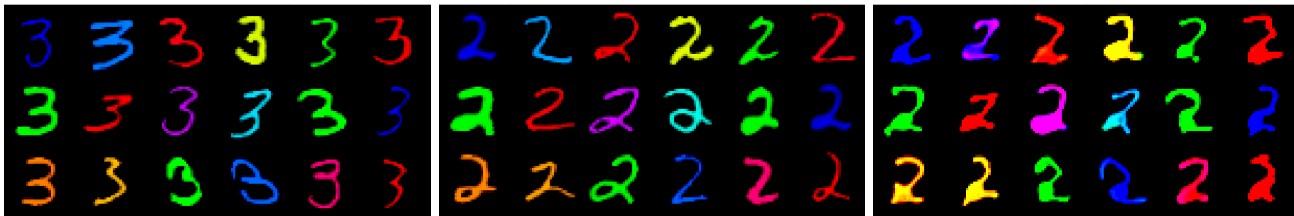

(a) Samples from $\mathbb{P}(\mathbf{x}^+)$, which are placed on the left plate $z = 0$.

(b) Samples from **our** approximation of $\mathbb{Q}(\mathbf{x}^-)$, located on the right plate $z = 10$.

(c) Samples from **FM**'s approximation of $\mathbb{Q}(\mathbf{x}^-)$, located on the right plate $z = 10$.

*Figure 10. Image-to-Image translation*. Pictures from the initial distribution, the result of applying our EFM method as well as the Flow Matching method are presented respectively.

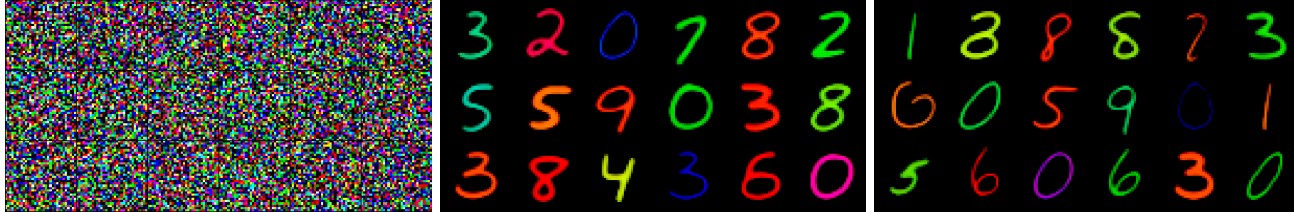

(a) White noise samples from $\mathbb{P}(\mathbf{x}^+)$, which are placed on the left plate $z = 0$.

(b) Samples from **our** approximation of $\mathbb{Q}(\mathbf{x}^-)$, located on the right plate $z = 30$.

(c) Samples from **PFGM**'s approximation of $\mathbb{Q}(\mathbf{x}^-)$, simulated from hemisphere with the learned field.

*Figure 11. Noise-to-Image generation*. Pictures from the initial distribution (Fig. 11a), the result of our EFM method (Fig. 11b) as well as the PFGM method (Fig. 11c) are presented.

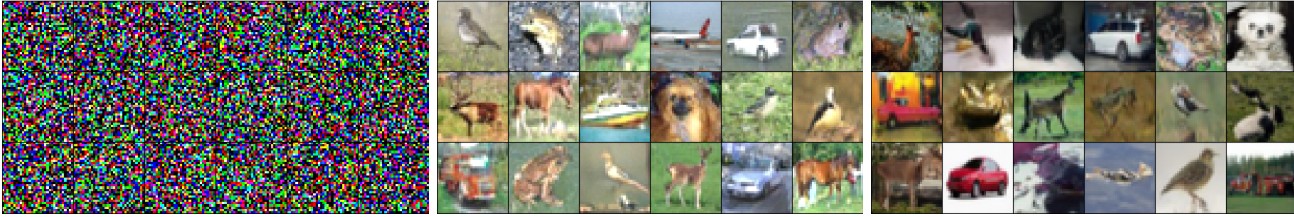

(a) White noise samples from $\mathbb{P}(\mathbf{x}^+)$ which are placed on the left plate $z = 0$.

(b) Samples from **our** approximation of $\mathbb{Q}(\mathbf{x}^-)$ located on the right plate $z = 500$.

(c) Samples from **PFGM**'s approximation of $\mathbb{Q}(\mathbf{x}^-)$ simulated from hemisphere with the learned field.

*Figure 12. Noise-to-Image generation*. Pictures from the initial distribution (Fig. 12a), the result of our EFM method (Fig. 12b) as well as the PFGM method (Fig. 12c) are presented.

accurately translate the shape and color of the initial digits 3. In Appendix D, we demonstrate the results of the translation obtained with several other relevant methods.

### 4.3. Image Generation Experiments

**MNIST**. We also consider the task of generating $32 \times 32$ full colored digits from the MNIST dataset. For this task, we place white noise on the left hyperplane $z = 0$ and colored digits on the right plate $z = 30$. After learning the electric field between the plates, we demonstrate mapping to the target distribution $\mathbb{Q}(\mathbf{x}^-)$ in Fig. 11b. We qualitatively see that even using only forward lines allows to recovers the target distribution $\mathbb{Q}(\mathbf{x}^-)$ of colored digits well. The sampling trajectories for our EFM are shown in Fig. 9.

**CIFAR-10**. The experiment with qualitative comparisons on more complex CIFAR-10 data is presented in Fig.12b. We place the white noise on the left plate $z = 0$, while images are placed on the right plate $z = 500$.

For completeness and comparison, we show the results of generation of (Xu et al., 2022, PFGM) which is also based on the electrostatic theory, see Fig.11c and Fig.12c.

## 5. Limitations

Our EFM method has several limitations which open promising avenues for future work and improvements.

**Influence of dimensionality**. In high dimensions, our algorithm may need to operate with small values. Specifically, the multiplier $1/||\mathbf{x} - \mathbf{x}'||^D$ in the electric field formula (9) can produce values comparable to machine precision as $D$ increases. As a result, the training of our method may become less stable. Note that the same holds for PFGM.

**The impact of interplate distance $L$ on the field.** The larger the interplate distance $L$ is, the more curved and disperse the electric field lines become, see, e.g., Fig. 8b. With an increase of the distance the electric field has to be accurately learned in a larger volume between the plates. A

careful selection of the hyperparameter $L$ may be important when applying our method. Further discussion of the choice of parameter $L$ can be found in Appendix C.1.

**Defining the optimal training volume.** Our training approach involves sampling points $\widetilde{\mathbf{x}}^+$ and $\widetilde{\mathbf{x}}^-$ from the distributions, interpolating them with (20) and noising them with (21). This allows us to consider an intermediate point $\widetilde{\mathbf{x}}$ between the plates (20) to learn the electrostatic field. There may exist more clever schemes to choose such points, see Appendix C.4. It is a promising question of further research.

**The problem of lines going beyond $z \in (0, L)$.** Modeling the movement along the electric field lines, e.g., forward stochastic map $T_F$ in (16), theoretically requires considering the lines which cross the boundary $z = L$. However, in practice, we ignore this requirement and stop when $z = L$, which may lead to incorrect learning of the target data distribution. We discuss this in Appendix C.2.

**Backward-oriented field lines.** Each plate emits two distinct field line sets: forward-oriented and backward-oriented trajectories. Our practical implementation exclusively utilizes forward-oriented lines due to their reduced training volume requirements. Nevertheless, backward-oriented trajectories remain theoretically significant, particularly for ensuring complete support coverage of $\mathbb{Q}(\mathbf{x}^-)$, see App. C.3.

## Impact Statement

This paper presents work whose goal is to advance the field of Machine Learning. There are many potential societal consequences of our work, none of which we feel must be specifically highlighted here.

## Acknowledgement

The work was supported by the grant for research centers in the field of AI provided by the Ministry of Economic Development of the Russian Federation in accordance with the agreement 000000C313925P4F0002 and the agreement with Skoltech №139-10-2025-033.

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

## A. Properties of D-dimensional electric field lines

In this auxiliary section, we formulate and prove the basic properties of electric field lines in $D$-dimensions.

**Definition A.1** (The flux of the electric field). The flux of the electric field $\mathbf{E}$ through an area $\mathbf{dS}$ is called $d\Phi = \mathbf{E} \cdot \mathbf{dS}$. The flux through a finite surface $\Sigma$ is defined as an integral:

$$\Phi = \int_\Sigma d\Phi = \iint_\Sigma \mathbf{E} \cdot \mathbf{dS}. \tag{24}$$

**Definition A.2** (The stream surface, Fig.13). Consider a closed piecewise smooth curve $\Gamma$ placed in an electric field. Consider field line passes through each point of this contour. The set of these lines is called a stream surface (tube).

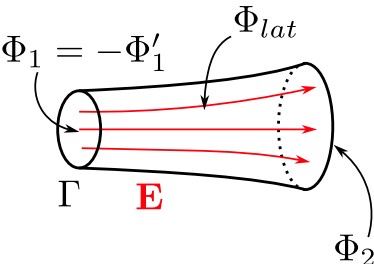

*Figure 13.* The electric field flux through an arbitrary stream tube.

**Lemma A.3** (Conservation of the field flux.). *The electric field flux is conserved along a stream surface if there are no charges inside that surface.*

*Proof.* Consider an arbitrary stream tube (Fig. 13). Let us cut this tube by two arbitrary piecewise smooth cross sections. As a result, we obtain a closed surface. Let us denote the flow through the side cross sections by $\Phi_1$ and $\Phi_2$, respectively. Note that the normal for closed surfaces is directed outwards. Near the right end of the tube, the normal and the electric field are co-directional, and near the left boundary, they have opposite directions. Therefore, $\Phi_1 = -\Phi_1'$. We have to prove that $\Phi_1' = \Phi_2$. The full flux is a sum of fluxes through the ends of the tube and through its lateral surface:

$$\Phi_{full} = \Phi_1 + \Phi_2 + \Phi_{lat}. \tag{25}$$

By the definition of a stream tube, the flux through the lateral surface must be zero: $\Phi_{lat} = 0$, because the field $\mathbf{E}$ and the normal to the lateral surface are orthogonal. Thus, $\Phi_{full} = \Phi_1 + \Phi_2 = \Phi_2 - \Phi_1'$. From Gauss's theorem (6), we derive:

$$\Phi_{full} = \iint_{\partial M} \mathbf{E} \cdot \mathbf{dS} = \int_M q(\mathbf{x})d\mathbf{x} = 0. \tag{26}$$

It follows that $\Phi_1' = \Phi_2$. $\qquad\square$

**Corollary A.4** (Impossibility of line termination in empty space). *An electric field line cannot terminate in empty space.*

*Proof.* Otherwise, the field flux inside the current tube surrounding the termination point will not be conserved. Intuitively, near a given point the field will enter, but will not exit, which is impossible in a region with a zero charge density. $\qquad\square$

**Lemma A.5** (The electric field flux from a point charge.). *Let us assume that a surface $\Sigma$ is seen at a solid angle[6] $\Omega$ from a point charge $q_0$ (Fig. 14). The electric field flux through this surface is equal to:*

---

[6]By definition, the solid angle $\Omega$ at which the given surface $\Sigma$ is visible from a given point Q is equal to the area of the projection of this surface $\Sigma$ onto the unit sphere centered at this point. Mathematically, the solid angle element in $\mathbb{R}^D$ is defined as $d\Omega_D = \frac{\mathbf{r} \cdot \mathbf{dS}}{r^D} = \frac{dS_\perp}{r^{D-1}}$, where $\mathbf{r} \in \mathbb{R}^D$ is a vector drawn from Q to the considered point on surface $\Sigma$, $r = ||\mathbf{r}||$, $dS_\perp = dS \cos \alpha = \mathbf{dS} \cdot \frac{\mathbf{r}}{r}$.

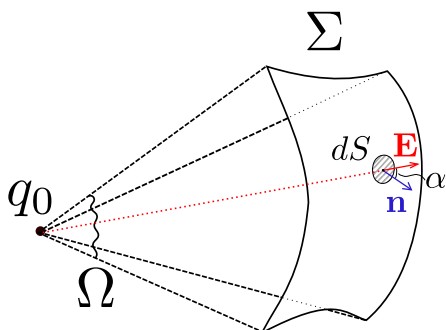

*Figure 14.* The field flux of a point charge $q_0$ through an arbitrary surface $\Sigma$ seen at solid angle $\Omega$.

$$\Phi = \frac{q_0 \Omega}{S_{D-1}}. \tag{27}$$

*Proof.* Divide $\Sigma$ into small surface elements $dS$. The total flux is the integral over the entire surface, $\Phi = \int_\Sigma d\Phi$. By definition of flux (Definition A.1), and according to (8) we have:

$$d\Phi = \mathbf{E} \cdot \mathbf{dS} = E dS \cos \alpha = E dS_\perp = \frac{q_0}{S_{D-1}} \frac{dS_\perp}{r^{D-1}} = \frac{q_0 d\Omega}{S_{D-1}}, \tag{28}$$

where $\alpha$ is the angle between $\mathbf{E}$ and $\mathbf{dS}$, $dS_\perp = dS \cos \alpha$. Then after integration over the solid angle, we obtain (27). $\quad\square$

**Lemma A.6** (Electric field lines start and end points). *Let an electrically neutral system, $\int q(\boldsymbol{x}) d\boldsymbol{x} = 0$, be bounded in space, i.e., $q(\boldsymbol{x})$ has compact support. Then the electric field lines must begin at positive charges and end at negative charges, except perhaps for the number of lines of zero flow.*

*Proof.* Let us denote the diameter of the system by $\ell = \max_{\mathbf{x}, \mathbf{x}' \in \text{supp}(q)}(||\mathbf{x} - \mathbf{x}'||)$. Consider our system at a large distance $L \gg \ell$. Let the small parameter $\xi = \ell/L \ll 1$ characterize this distance. We use the multipole decomposition of an electric field (Landau & Lifshitz, 1971, §40-41):

$$\mathbf{E}|_{r \to L} = \mathbf{E}^{(0)} + \mathbf{E}^{(1)} + \dots = \mathbf{E}_{\text{point}} + O\left(\frac{\xi}{L^{D-2}}\right). \tag{29}$$

Multipole decomposition is a representation of an electric field as a sum of contributions from point sources of different multipoles, monopole $\mathbf{E}^{(0)} = \mathbf{E}_{\text{point}}$, dipole $\mathbf{E}^{(1)}$, quadrupole $\mathbf{E}^{(2)}$, etc. It allows describing the field at large distances from the system, simplifying the calculations by neglecting the contributions of higher multipoles. In our problem, the multipole decomposition shows that the field at a large distance from the system looks like a field of a point charge located at the origin. All other contributions can be neglected. And therefore, in the limiting case $\xi \to 0, L \to \infty$ the formula (27) can be used. Since there is no limit on the increase of $L$, one can achieve an approximation accuracy as high as one needs.

Then $\Phi = \int \mathbf{E} \cdot \mathbf{dS} = Q\Omega/S_{D-1} = 0$, because $Q = \int q(\mathbf{x}) d\mathbf{x} = 0$ is the total charge. Hence, the flux through any surface located at a distance $||\mathbf{r}|| \geq L$ from the system is zero:

$$\int \mathbf{E} \cdot \mathbf{dS} = 0 \text{ when } ||\mathbf{r}|| \geq L \to \infty. \tag{30}$$

Suppose a non-measure-zero set of field lines escapes to infinity. These lines would generate flux $\Phi \neq 0$ in contradiction to $\Phi = 0$. Thus, all field lines originate at positive charges and terminate at negative ones, except for a set of measure zero. $\quad\square$

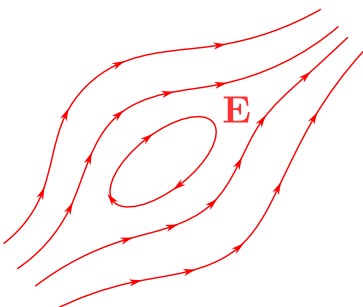

*Figure 15.* Closed loop of an electric field line. This situation is impossible due to the circulation theorem.

**Lemma A.7** (Absence of closed electric field loops). *Electric field lines cannot form closed loops (as shown in Fig. 15).*

*Proof.* Assume, to the contrary, that a closed loop $\ell$ exists. Then $\mathbf{E} \cdot d\mathbf{l} \geq 0$ holds everywhere along $\ell$ (since $\mathbf{E}$ and $d\mathbf{l}$ are co-directed), and the equality to zero of $\mathbf{E} \cdot d\mathbf{l} = 0$ cannot be satisfied everywhere (otherwise it is not a closed loop and even not a field line). Consequently, the circulation integral satisfies:

$$\oint_\ell \mathbf{E} \cdot d\mathbf{l} > 0.$$

However, the circulation theorem (7) requires:

$$\oint_\ell \mathbf{E} \cdot d\mathbf{l} = 0,$$

yielding a contradiction. $\qquad \square$

## B. Proof of Electrostatic Field Matching theorem

In this section, we prove several auxiliary lemmas, and then move on to the main theorem. Let $\mathbb{P}(\cdot)$, $\mathbb{Q}(\cdot)$ be two $D$-dimensional data distributions having a compact support, located on the planes $z = 0$ and $z = L$ in $\mathbb{R}^{D+1}$, respectively.

**Lemma B.1** (On the relation between distribution and field values). *For any point in the support of distribution $\mathbb{P}(\cdot)$:*

$$E_z^+(\widetilde{\boldsymbol{x}}) - E_z^-(\widetilde{\boldsymbol{x}}) = \mathbb{P}(\widetilde{\boldsymbol{x}}). \tag{31}$$

*Proof.* Consider an infinitesimal volume $\mathbf{dS}$ is the support of $\mathbb{P}$. Consider a closed surface, a cylinder with infinitesimal indentation in different directions in the plane $z = 0$, see Fig. 16.

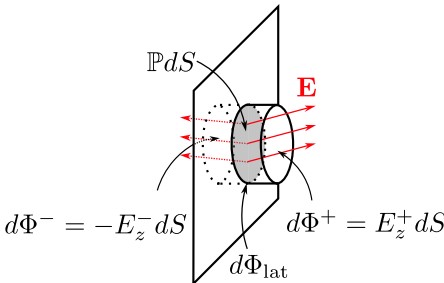

*Figure 16.* The considered area.

The flux through this surface consists of three summands: $d\Phi^+$, the flux in the positive direction of the $z$-axis; $d\Phi^-$, the flux in the negative direction of the $z$-axis; and $d\Phi_{\text{lat}}$, the flux through the lateral surface:

$$d\Phi_{\text{full}} = d\Phi^+ + d\Phi^- + d\Phi_{\text{lat}} = E_z^+ dS - E_z^- dS + 0. \tag{32}$$

Here $d\Phi_{\text{lat}} = 0$ since the height of the cylinder under consideration can be made as small as one wants (infinitesimal of higher order than $dS$). In turn, $d\Phi^- = -E_z^- dS$ has a negative sign due to the fact that the normal to the closed surface is directed outward, i.e. in the opposite direction from the axis $z$. Then, due to the Gauss's theorem (6):

$$d\Phi_{\text{full}} = (E_z^+ - E_z^-)dS = \mathbb{P}dS \tag{33}$$

which proves the lemma. □

Let $\{\mathbf{x}_i^+\}_{i=1}^n$ be a sample of points distributed over $\mathbb{P}$. Let $dn$ be the number of points from the sample that fall in the volume $dS$ in the support of $\mathbb{P}$. Then $dn = dn_F + dn_B$, where $dn_F$ is the number of points in $dS$ that correspond to the mapping $T_F$ (that is, the movement along forward oriented lines; see (17)), and $dn_B$ corresponds to $T_B$.

**Lemma B.2** (First lemma on the flow). *Let the field near the element $dS$ have different signs on different sides: $E_z^+ > 0$ and $E_z^- < 0$. Then:*

$$\begin{aligned}
\frac{dn_F}{n} &\xrightarrow[n\to\infty]{a.s.} E_z^+ dS = d\Phi_F, \\
\frac{dn_B}{n} &\xrightarrow[n\to\infty]{a.s.} |E_z^-|dS = d\Phi_B,
\end{aligned} \tag{34}$$

*where $\left(\ldots \xrightarrow[n\to\infty]{a.s.} \ldots\right)$ denotes the almost sure convergence, and $d\Phi_{F,B}$ denotes the electric field flux near the plate surface in the forward and backward directions.*

*Proof.* According to the multiplication rule of probability and the law of large numbers:

$$\begin{aligned}
\frac{dn_F}{n} &\xrightarrow[n\to\infty]{a.s.} (\text{probability of choosing } T_F) \cdot (\text{probability of falling in } dS) = \\
&= \mu(\widetilde{\mathbf{x}}) \cdot \mathbb{P}(\widetilde{\mathbf{x}})dS = \frac{E_z^+}{E_z^+ + |E_z^-|} \cdot (E_z^+ + |E_z^-|)dS = E_z^+ dS = d\Phi_F.
\end{aligned} \tag{35}$$

In the second equality, the definitions of probability $\mu(\cdot)$, see (18), and Lemma B.1 are used. The case $dn_B$ is proved similarly. □

**Lemma B.3** (Second lemma on the flow). *Let $\mathbb{P}, \mathbb{Q}, \{\mathbf{x}_i^+\}_{i=1}^n, dS, dn$ have the same meaning as in Lemma B.2. Let $E_z^+$ and $E_z^-$ have the same sign near $dS$ (i.e., $E_z^\pm > 0$). Then*

$$\frac{dn}{n} \xrightarrow[n\to\infty]{a.s.} d\Phi_{after} - d\Phi_{before}, \tag{36}$$

*where $d\Phi_{before}$ is the field flux through the current tube supported on $dS$ immediately before crossing the plane $dS \in \text{supp}\,\mathbb{P}(\cdot)$, and $d\Phi_{after}$ is the flux after crossing.*

*Proof.* By the probability product theorem, the strong law of large numbers, Lemma B.1, and the definition of flux:

$$\frac{dn}{n} \xrightarrow[n\to\infty]{a.s.} \mu(\mathbf{x})\mathbb{P}(\mathbf{x})dS = 1 \cdot \mathbb{P}(\mathbf{x})dS = (E_z^+ - E_z^-)dS = d\Phi_{after} - d\Phi_{before}. \tag{37}$$

□

**Remark.** This statement implies that when the field crosses the plane $\mathbb{P}$ containing a charge (proportional to $dn/n$), the field flux must *increase* by $dn/n$.

Lemmas B.2 and B.3 address the behavior near the distribution $\mathbb{P}$. Similar statements are valid for the behavior near $\mathbb{Q}$. When moving along field lines, we eventually reach the plane $z = L$. At this point $\widetilde{\mathbf{x}}^-$, two different scenarios may occur:

1. $E_z^+(\widetilde{\mathbf{x}}^-)$ and $E_z^-(\widetilde{\mathbf{x}}^-)$ have opposite signs. Then the field line motion terminates in this case.

2. $E_z^+(\widetilde{\mathbf{x}}^-)$ and $E_z^-(\widetilde{\mathbf{x}}^-)$ have the same sign. Then some number $dn' \leq dn$ of lines inside current tube (corresponding to sample $\{\mathbf{x}_i^+\}_{i=1}^n \sim \mathbb{P}$) must terminate, while the others should continue moving.

This portion $dn'$ can be found from the line termination property in $\mathbb{Q}$.

**Lemma B.4** (Line Termination). *If $E_z^+(\widetilde{\mathbf{x}}^-)$ and $E_z^-(\widetilde{\mathbf{x}}^-)$ have the same sign upon crossing $z = L$, the number of lines terminating on $z = L$ satisfies:*

$$\frac{dn'}{n} \xrightarrow[n\to\infty]{a.s.} -d\Phi_{after} + d\Phi_{before}, \tag{38}$$

**Remark.** When the field crosses the plane $\mathbb{Q}$ containing a charge (proportional to $dn'/n$), the field flux must *decrease* by $dn'/n$.

*Proof.* Consider the current tube before it intersects the plane $z = L$. Let us denote the number of lines inside $dn_{\text{before}}$. As a result of the intersection $z = L$, some of the lines $dn'$ stop moving, while some of the lines $dn_{\text{after}}$ continue moving. In view of the first Lemma B.2 on flow, as well as the conservation of flow inside the current tube (Lemma A.3):

$$\frac{dn_{\text{before}}}{n} \xrightarrow[n\to\infty]{a.s.} d\Phi_{\text{before}}. \tag{39}$$

Then, by virtue of the law of large numbers and the fact that $dn_{\text{before}} = dn' + dn_{\text{after}}$, we have:

$$\frac{dn'}{n} \xrightarrow[n\to\infty]{a.s.} (\text{probability of termination}) \cdot \frac{dn_{\text{before}}}{n} \xrightarrow[n\to\infty]{a.s.} \nu(\mathbf{x}^-) \cdot d\Phi_{\text{before}}$$
$$\frac{E_z^- - E_z^+}{E_z^-} \cdot E_z^- dS' = (E_z^- - E_z^+)dS' = -d\Phi_{\text{after}} + d\Phi_{\text{before}} \tag{40}$$

$\square$

We now proceed to prove the main theorem.

**Theorem B.5** (Electrostatic Field Matching). *Let $\mathbb{P}(\boldsymbol{x}^+)$ and $\mathbb{Q}(\boldsymbol{x}^-)$ be two data distributions that have compact support. Let $\boldsymbol{x}^+$ be distributed as $\mathbb{P}(\boldsymbol{x}^+)$. Then $\boldsymbol{x}^- = T(\boldsymbol{x}^+)$ is distributed as $\mathbb{Q}(\boldsymbol{x}^-)$ almost surely:*

$$\text{If } \boldsymbol{x}^+ \sim \mathbb{P}(\boldsymbol{x}^+) \Rightarrow T(\boldsymbol{x}^+) = \boldsymbol{x}^- \sim \mathbb{Q}(\boldsymbol{x}^-). \tag{41}$$

*Proof.* 1. Let $\{\mathbf{x}_i^+\}_{i=1}^n$ be a sample of points distributed according to $\mathbb{P}(\cdot)$. Applying the mapping $T(\cdot)$, we obtain the corresponding points on the target distribution: $\{\mathbf{x}_i^-\}_{i=1}^n = \{T(\mathbf{x}_i^+)\}_{i=1}^n$.

2. Consider a $D$-dimensional area element $dS'$ on $\mathbb{Q}(\cdot)$. Let $dn'$ denote the number of points $\mathbf{x}_i^-$ within this region. We define the empirical distribution:

$$\hat{\mathbb{Q}}_n(\mathbf{x}^-)dS' = \frac{dn'}{n}. \tag{42}$$

Our goal is to prove:

$$\hat{\mathbb{Q}}_n(\cdot) \xrightarrow[n\to\infty]{\text{almost surely}} \mathbb{Q}(\cdot). \tag{43}$$

3. The $dn'$ points arrive at $\mathbb{Q}(\cdot)$ through two pathways (Fig. 17), forward-oriented ($dn'_F$) and backward-oriented ($dn'_B$) trajectories:

$$dn' = dn'_F + dn'_B. \tag{44}$$

4. Consider the current tube terminating at $dS'$ corresponding to forward-oriented arrivals ($dn'_F$). Since $\mathbb{P}(\cdot)$ and $\mathbb{Q}(\cdot)$ have compact supports and the total charge related to the system of the two plates is zero, by Lemma A.6, these tubes must start from $\mathbb{P}(\cdot)$ almost surely.

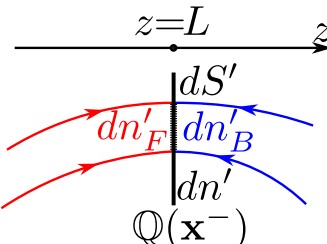

*Figure 17.* An element of volume $dS'$ is selected on the distribution $\mathbb{Q}(\cdot)$. $dn'$ is the number of points $x_i^-$ falling into this volume. Some points came to the distribution $\mathbb{Q}(\cdot)$ from the front (denote them by $dn'_F$), and some came from the back ($dn'_B$).

5.  During the motion against field lines from $\mathbb{Q}$ to $\mathbb{P}$, multiple crossings of $z = 0$ and/or $z = L$ may occur ($N = 0, 1$ or $2$, see Fig. 18). Denote the intersection points:

$$\mathbf{x}^- = \mathbf{x}_0 \rightarrow \mathbf{x}_1 \rightarrow \cdots \rightarrow \mathbf{x}_N \rightarrow \mathbf{x}_{N+1} = \mathbf{x}^+. \tag{45}$$

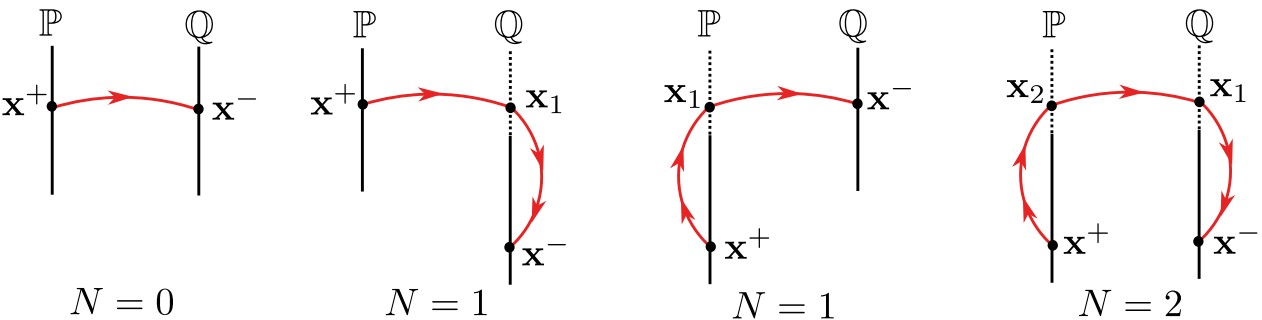

*Figure 18.* Intersection points.

Their corresponding area elements are

$$dS' = dS_0 \rightarrow dS_1 \rightarrow \cdots \rightarrow dS_N \rightarrow dS_{N+1} = dS. \tag{46}$$

Point counts in these areas are

$$dn' = dn_0 \rightarrow dn_1 \rightarrow \cdots \rightarrow dn_N \rightarrow dn_{N+1} = dn, \tag{47}$$

where $dn_k$ ($k = 0, ..., N + 1$) is number of points from sample $\{\mathbf{x}_i\}_{i=1}^n$ or from map $\{T(\mathbf{x}_i)\}_{i=1}^n$ inside the volume $dS_k$ near point $\mathbf{x}_k$ that corresponds to considered motion inside current tube.

6.  The $dn_k$ are not arbitrary but related by flux conservation. Only the charged planes ($z = 0$ or $z = L$) can alter the count:

    - At $z_k = 0$: Line count increases by $dn_k$.
    - At $z_k = L$: Line count decreases by $dn_k$.

It can be written mathematically as:

$$\sum_{k=0}^{N+1} (-1)^{f_k} dn_k = 0, \tag{48}$$

where

$$f_k = \begin{cases} 0 & \text{if } z_k = 0, \\ 1 & \text{if } z_k = L. \end{cases} \tag{49}$$

7. Due to the first Lemma B.2 on flow:

$$\frac{dn_{N+1}}{n} \equiv \frac{dn}{n} \xrightarrow[n\to\infty]{\text{a.s.}} d\Phi_{N+1} \equiv d\Phi, \tag{50}$$

8. Due to the second Lemma B.3 on the flow, and because of the line termination Lemma B.4:

$$(-1)^{f_k} \cdot \frac{dn_k}{n} \xrightarrow[n\to\infty]{\text{a.s.}} d\Phi_{\text{after},k} - d\Phi_{\text{before},k}. \tag{51}$$

9. According to the law of conservation of flux along the tube (Lemma A.3) from $\mathbf{x}_i$ to $\mathbf{x}_{i-1}$:

$$d\Phi_{\text{after},k} = d\Phi_{\text{before},k-1}. \tag{52}$$

10. Whence we obtain a chain of equalities:

$$0 = \sum_{k=0}^{N+1} (-1)^{f_k} dn_k = -\frac{dn'_F}{n} + (-1)^{f_1} \frac{dn_1}{n} + ... + (-1)^{f_N} \frac{dn_N}{n} + \frac{dn}{n} \Rightarrow$$

$$\frac{dn'_F}{n} = (-1)^{f_1} \frac{dn_1}{n} + ... + (-1)^{f_N} \frac{dn_N}{n} + \frac{dn}{n} \xrightarrow[n\to\infty]{\text{a.s.}} \tag{53}$$

$$\xrightarrow[n\to\infty]{\text{a.s.}} -d\Phi_{\text{after},1} - d\Phi_{\text{before},1} + ... + d\Phi_{\text{after},N} - d\Phi_{\text{before},N} + d\Phi_{N+1} =$$

$$= d\Phi_{\text{after},1} + 0 + ... + 0 = d\Phi'_F.$$

Consequently,

$$\frac{dn'_F}{n} \xrightarrow[n\to\infty]{\text{a.s.}} d\Phi'_F. \tag{54}$$

Note that this property is not obvious in general, since the points $dn'$ are obtained by applying the mapping $T(\cdot)$ to a certain *sample* of points, while the flux $d\Phi'$ is determined by the *true* value of the field, which is defined by the entire distribution of charges $\mathbb{P}$ and $\mathbb{Q}$.

Similarly, it can be proven that

$$\frac{dn'_B}{n} \xrightarrow[n\to\infty]{\text{a.s.}} d\Phi'_B. \tag{55}$$

11. Then, by virtue of the Gauss's theorem (6), we finally have

$$\hat{\mathbb{Q}}_n dS' = \frac{dn'}{n} = \frac{dn'_F}{n} + \frac{dn'_B}{n} \xrightarrow[n\to\infty]{\text{a.s.}} \frac{d\Phi'_F}{\Phi_0} + \frac{d\Phi'_B}{\Phi_0} = \mathbb{Q}dS. \tag{56}$$

This completes the proof.

$\square$

## C. Extended discussion of limitations

### C.1. The problem of choosing the hyperparameter $L$

The interplate distance L between the left and right plates of the capacitor is the key tuning hyperparameter that influences the performance. In the §4.1, we provide an experiment on Swiss roll dataset that demonstrates the impact of L on the performance (Fig. 8). With neural-network-based approaches, the electric field becomes less recoverable as the hyperparameter increases as, informally speaking, the required training volume also grows.

We also carry out another example to show the influence of the hyperparameter $L$ in noise-to-image generation CIFAR-10 experiment (Fig. 19). It can be seen that increasing $L$ up to the value of $5000$ leads to a significant decrease in the generation quality. At the same time, no significant differences are observed for the average values of $L = 50, 500$. The theoretical ideal $L^\star$ that minimizes the approximation error remains unknown and represents an important direction for future work. Hovewer, usually we choose $L$ of an order comparable to the standard deviation of the data distributions.

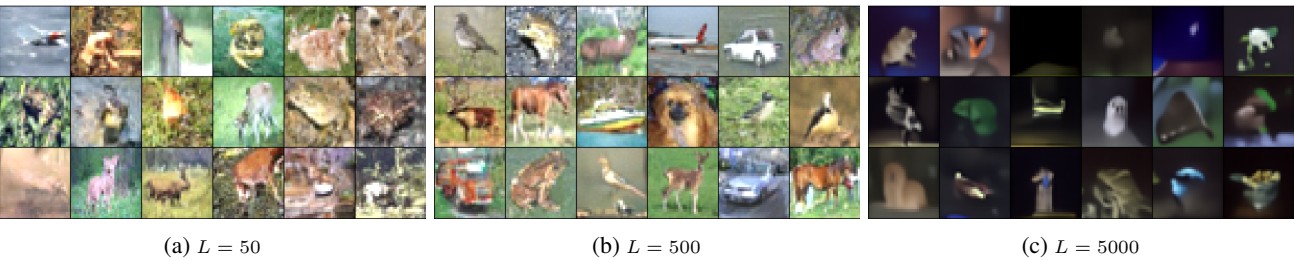

(a) $L = 50$        (b) $L = 500$        (c) $L = 5000$

*Figure 19. The influence of the **interplate distance** $L$ on the performance of the Noise-to-Image generation of CIFAR-10 dataset. Pictures EFM method with $L = 50$ (Fig. 19a), $L = 500$ (Fig. 19b) as well as $L = 5000$ (Fig. 19c) are presented.*

### C.2. The problem of lines going beyond $z = L$

When moving from $\mathbb{P}$ to $\mathbb{Q}$ along the field lines, we eventually reach the plane $z = L$. Two different scenarios may occur:

1. $E_z^+(L)$ and $E_z^-(L)$ have opposite signs. The field line motion terminates in this case.

2. $E_z^+(L)$ and $E_z^-(L)$ have the same sign. Then, the movement should end at this point with probability $\nu(\mathbf{x}^-)$, and with probability $1 - \nu(\mathbf{x}^-)$, the movement should continue (recall (15)).

Overall, one needs a stochastic choice for terminating and/or continuing the movement. To make this choice, it is necessary to calculate the value of the field $E_z^\pm(L)$ to the left and right of the plate $z = L$ at the point of interest and calculate $\nu(\mathbf{x}^-)$.

Suppose that the situation of continuing movement is realized. This motion should be performed by integrating $d\mathbf{x}(t) = \mathbf{E}(\mathbf{x}(t))dt$ until the line is on the target distribution again. Note that such a motion should indeed end on the distribution $\mathbb{Q}$ (Lemma A.6), and also that further movement is impossible (since $E_z^+(L) < 0$, and between plates $z \in (0, L)$ we always have $E_z > 0$, in particular $E_z^-(L) > 0$, i.e., the first situation is realized). An example of considered problem is shown in Fig. 20. Note that in practice, the field is calculated from a sample of data (rather than from a continuous distribution), so it is difficult to calculate $E_z^\pm$ accurately. Therefore, in practice we always stop the movement at the first intersection.

### C.3. Backward-oriented field lines

Transport between plates can occur via two distinct trajectory classes (Fig. 21a):

- **Forward-oriented trajectories** (red lines): Initial motion directed toward the target plate

- **Backward-oriented trajectories** (black lines): Initial motion receding from the target plate

Note that due to the Lemma A.6, both series must end at the second distribution. The difference between the two series is the greater curvature of the backward-oriented series compared to the forward-oriented one. From the practical point of view,

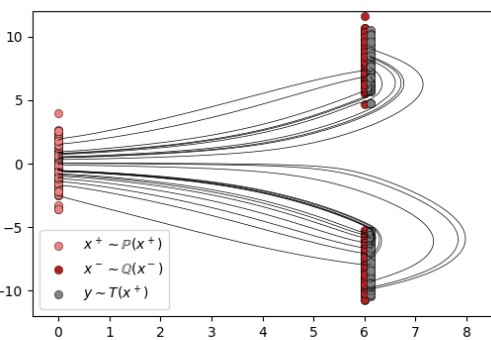

*Figure 20.* A toy experiment Gaussian→ 2Gaussians. The two Gaussians are significantly separated from each other, so a large number of lines flying out of the region $z = L$ is observed. Further integration along the field lines still leads to the target distribution.

using backward-oriented series is less useful than forward-oriented. This is not only due to the curvature of the lines, but also due to the significantly larger training volume ($V_{\text{backward}}$ shaded in black in the figure is much larger than red $V_{\text{forward}}$).

We also note another feature related to the training volume. In Fig. 21a, we can see that lines starting at the periphery of the data distribution are particularly curved, and may partially leave the training volume (points $A, B, C, D, E, F, G, H$).

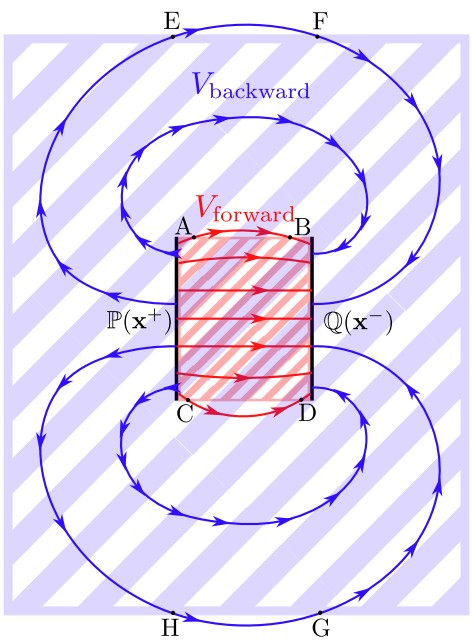

(a) Two series of field lines: Forward-directed (red lines), and backward-directed (blue lines). Blue lines require much larger training volume $V_{\text{backward}}$ than red lines, which take up $V_{\text{forward}}$. The figure also shows the field lines starting from the peripheral distribution points that can leave the training volume (points $A, B, C, D, E, F, G, H$).

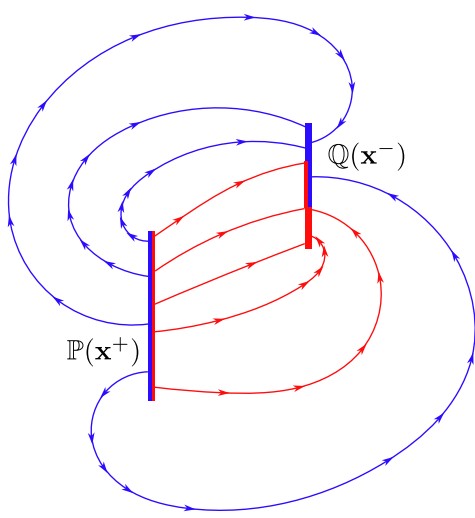

(b) Two series of field lines: forward-directed (red lines), and backward-directed (blue lines). It can be seen that forward-oriented (red) lines starting at the distribution $\mathbb{P}(\cdot)$ may not completely cover the target distribution $\mathbb{Q}(\cdot)$.

*Figure 21.* An illustration of forward-oriented and backward-oriented lines.

When source ($\mathbb{P}$) and target ($\mathbb{Q}$) distributions exhibit significant mean shifts ($\mu_{\mathbb{P}} \neq \mu_{\mathbb{Q}}$), the number of forward-oriented lines extending beyond the $z = L$ boundary can be quite large. In such a situation, there appears a region on the second distribution

that is not covered by forward-oriented lines (in Fig. 21b red lines do not completely cover the second distribution).

### C.4. Training Volume Selection

Our electrostatic framework allows flexible training volume definitions between plates. The (20) formula is not the only possible training volume option (different from the flow matching, which uses this particular interpolant for loss construction). In Fig. 22, a comparison of the two approaches is shown:

- Linear interpolation between samples of $\mathbb{P}$ and $\mathbb{Q}$ using (20)

- Uniform cube mesh between plates.

Figure 22 showcases the equivalent performance for both approaches in Gaussian-to-Swiss-roll transport.

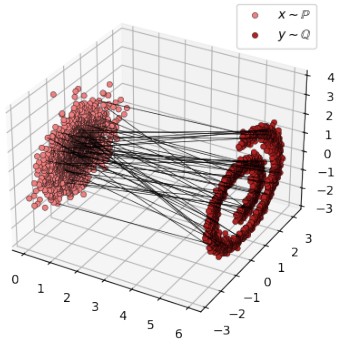

(a) Training volume for **our** EFM method that is defined by interpolation from (20).

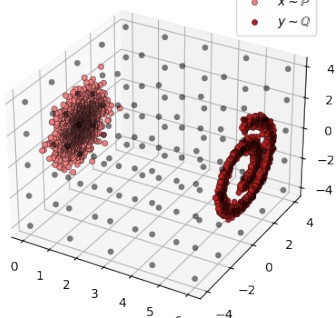

(b) Training volume for **our** EFM method that is defined by uniform cube mesh between plates.

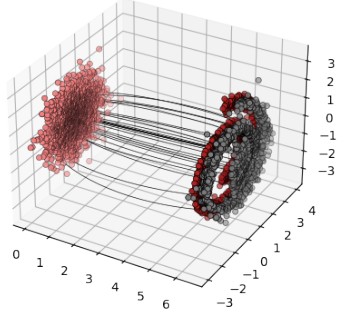

(c) The learned trajectories of **our** EFM with training volume defined in 20.

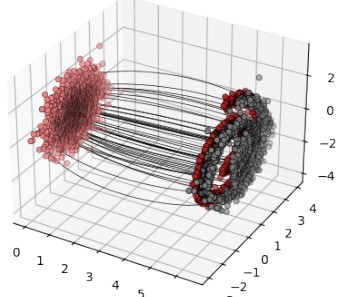

(d) The learned trajectories of **our** EFM with training volume defined by the cube mesh.

*Figure 22.* Learned trajectories of **our** EFM method with different training volumes on *Gaussian-to-Swiss roll experiment*.

## D. Experimental details

We aggregate the hyper-parameters of our Algorithm 1 for different experiments in the Table 1. We base our code for the experiments on PFGM's code `https://github.com/Newbeeer/Poisson_flow`.

**Toy experiments.** In the 2D illustrative example (§4.1), we make the inference by constructing the ODE Euler solver for the equation 11 with the iterative scheme (see Alg.2)

**Image data experiments.** In the case of the Image experiments (§4.2 and §4.3), we use the RK45 ODE solver provided by `https://docs.scipy.org/doc/scipy/reference/generated/scipy.integrate.RK45.html` for the inference process with the hyper-parameters rtol=1e-4 , atol=1e-4 and number of steps (NFE) equals to 100. Also, we use Exponential Moving Averaging (EMA) technique with the ema rate decay equals to 0.99 . As for the optimization procedure, we use Adam optimizer (Kingma & Ba, 2015) with the learning rate $\lambda = 2e-4$ and weight decay equals to 1e-4.

Evaluation of the training time for our solver on the image's experiments (see §4.2 and §4.3)takes less than 10 hours on a single GPU GTX 1080ti (11 GB VRAM).

| Experiment | D | Batch Size | $L$ | $NFE$,Num Steps | $\lambda$,LR | Weight Decay | $\sigma$ |
|---|---|---|---|---|---|---|---|
| Gaussian Swiss-roll | 2 | 1024 | 6 | 20 | 2e-3 | 0. | 0.001 |
| Colored MNIST Translation (3→2) | 3072 | 64 | 10 | 100 | 2e-4 | 1e-4 | 0.01 |
| Colored MNIST Generation | 3072 | 64 | 30 | 100 | 2e-4 | 1e-4 | 0.01 |
| CIFAR-10 Generation | 3072 | 64 | 500 | 100 | 2e-4 | 1e-4 | 0.05 |

*Table 1.* Hyper-parameters of Algorithm 1 for different experiments, where $D$ is the dimensionality of task, $L$ is the distance betwenn plates and $\sigma$ is used for the definition points between plates (see §3.3).

**Baselines.** We use the source code `https://github.com/Newbeeer/pfgmpp` for running **PFGM** in our experiments. We found the following values of hyper parameters are appropriate for us: $\gamma = 5, t = 0.3, \varepsilon = 1e-3$, see (Xu et al., 2022) for details. Also, we utilize the source code of Flow Matching (**FM**) from the github page `https://github.com/atong01/conditional-flow-matching/tree/main` for experiments in §4.2. We also add extra baselines: **Cycle-GAN** from the github page `https://github.com/junyanz/pytorch-CycleGAN-and-pix2pix`, **DDIB** from `https://github.com/suxuann/ddib` and **DSBM** from `https://github.com/yuyang-shi/dsbm-pytorch`. Their results are shown in Figure 23.

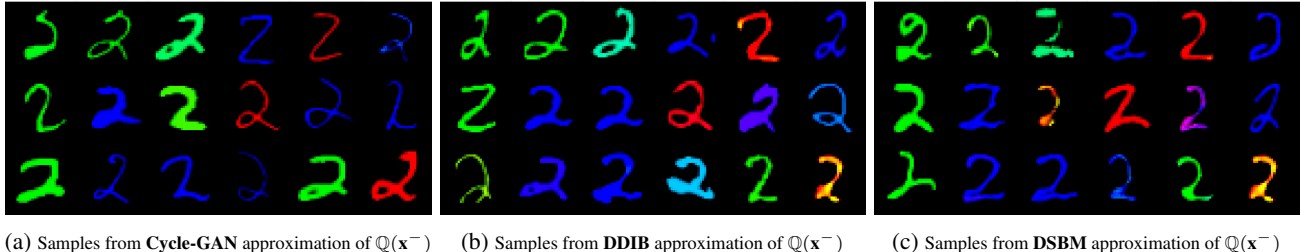

(a) Samples from **Cycle-GAN** approximation of $\mathbb{Q}(\mathbf{x}^-)$    (b) Samples from **DDIB** approximation of $\mathbb{Q}(\mathbf{x}^-)$    (c) Samples from **DSBM** approximation of $\mathbb{Q}(\mathbf{x}^-)$

*Figure 23. Image-to-Image translation* experiment (Colored MNIST dataset, $3 \to 2$). The results of alternative translation methods: **Cycle-GAN** (Zhu et al., 2017), **DDIB** (Su et al., 2023) and **DSBM** (De Bortoli et al., 2024)

