# OpenReview forum: "Field Matching: an Electrostatic Paradigm to Generate and Transfer Data"
_ICML.cc/2025/Conference — ICML 2025 poster_

### Official Review · Reviewer_YMMk · 2025-03-14

**Overall Recommendation:** 2

**Summary:**

The paper proposed a mechanism to train generative models. The main idea is to regard each data sample as a charge. The training process involves training networks to predict fields (gradients of potentials). The sampling process is done by solving an ordinary differential equation (moving data samples along fields).

**Claims And Evidence:**

Yes

**Essential References Not Discussed:**

Yes

**Experimental Designs Or Analyses:**

Considering the similarity between the paper and flow matching, I would like to know about some ablation studies.

1. Is the term in Eq 19 important? In flow matching's training, we do not need the term. I still do not know if it is necessary to make the design work.
2. The results are weak. Only some visual comparisons with PFGM are provided. However, a comparison with RF/FM is necessary.

**Methods And Evaluation Criteria:**

Only toy datasets like 2D point sets and MNIST are provided.

**Other Comments Or Suggestions:**

After reading the paper, I have no interest in using the method in my projects. The designs are similar to rectified flows but more complicated. If the authors can ablate the designs, I would be more interested in the paper.

**Other Strengths And Weaknesses:**

No

**Questions For Authors:**

No

**Relation To Broader Scientific Literature:**

The method is inspired by fields in physics. However, the method is just a complicated version of rectified flow. It is hard for me to judge the quality of the method based on the current draft.

**Theoretical Claims:**

The idea sounds interesting at first sight. However, when I review the training algorithm, they looked very similar tthe o existing diffusion model design.

1. The interpolation in Eq 19 is exactly the one used in rectified flow and flow matching (RF/FM), except for the random noise term.
2. Eq 11 and Eq 16 are used for the target truth in training the network. The term in Eq 11 involves a term (x-x') which is also the one used in RF/FM, except for the normalizing term. Thus the training loss is just a scaled version of RF/FM.

---

> ### Author Rebuttal · Authors · 2025-04-01
>
> Dear reviewer, thank you for  reviewing our paper. Below we answer your  questions and comments.
>
> **(Q1) Only some visual comparisons with PFGM are provided. However, a comparison with RF/FM is necessary. Only toy datasets like 2D point sets and MNIST are provided.**
>
> It is worth noting that our main goal is to demonstrate proof-of-concept of our method in the experimental section. We agree that providing additional generating experiments with more complex datasets enhances the understanding of the method's performance. Further scalability of the method is a promising avenue for future research.
>
> Nevertheless following your request, we include $\textbf{additional experiments}$ with more complex data such as CIFAR-10. For qualitative analysis, we demonstrate our EFM's results as well as PFGM's. We would like to ask you to get acquainted with Fig. 3 that is available via the anonymous link https://drive.google.com/file/d/1DTbQR_GNah7hVGGnDF822aD96iWxjR-k/view?usp=sharing.
>
> For quantitative analysis, we calculate FID/CMMD scores on test part of the aforementioned datasets and compare our method with PFGM, DDPM, DDIM, GLOW. Firstly, we demonstrate quantitative performance of our method on full Colored MNIST dataset. We see that our method outperforms other approaches on full Colored MNIST, reaching the lowest FID/CMMD.
>
> | Metrics/Method | EFM (our) | PFGM | DDPM | DDIM | GLOW |
> |----------------|----------|------|------|------|------|
> | FID            | **0.92** | 1.88 | 2.18 | 2.23 | 25.9 |
> | CMMD           | **1.47** | 2.28 | 2.68 | 2.85 | -    |
>
> Secondly, we demonstrate quantitative performance of our method on CIFAR-10 dataset. We see that in image generation on CIFAR-10, our performance is comparable to PFGM. *At the same time, we remind that our method is also capable of performing data-to-data transfer while PFGM is not capable of doing that*.
>
> | Metrics/Method | EFM (our) | PFGM       | DDPM | DDIM | GLOW  |
> |----------------|-----------|------------|------|------|-------|
> | FID            | 2.62      | **2.48**   | 3.17 | 4.16 | 48.9  |
> | CMMD           | **1.87**  | 1.93       | 2.98 | 3.25 | -     |
>
> In accordance of your request of comparison with  FM, we quantitatively and qualitatively demonstrate the performance of our approach with SB-based(DDIB, $\alpha$-DSBM ) approaches as well as GAN-based (Cycle-GAN ) for unpaired data setups on Colored MNIST. Please, familiarize yourself with Fig. 6 via the afore mentioned link. For quantitative analysis, we demonstrate FID and CMMD metrics on Colored MNIST dataset for our and compared methods.
>
> | Metrics/Method | EFM (our) | FM    | DSBM | DDIB | GAN  |
> |----------------|-----------|-------|------|------|------|
> | FID            | **4.45**  | 19.87 | 7.21 | 8.24 | 4.57 |
> | CMMD           | **2.37**  | -     | 4.02 | 4.11 | 2.45 |
>
> We see that our method demonstrates the highest FID and CMMD among compared approaches.
>
> **(Q2)  [...] rectified flow and flow matching (RF/FM) [...] the training loss is just a scaled version of RF/FM.**
>
> We think that there might be a misunderstanding of our EFM method. In fact, we do not have a lot in common with flow matching (FM), except the fact that we learn an ODE to generate or transfer data. In particular, all the theoretical derivations and motivation totally differ.
>
> - We work in $D+1$ dimensional space and learn a static (non time-dependent) vector field to transfer data, while the field in FM is time-dependent and is in $D$-dimensional space.
>
> - The FM defines the interpolation $tx + (1-t)y$ between data samples $x$ and $y$ from two different data sets to regress a velocity field on $y-x$, where $t$ is sampled from a standard uniform distribution. In fact, the usage of this particular interpolant is principle for their loss construction. In turn, in our case, this interpolation is just a $\textit{technical}$ way to define some inter-plate points $\widetilde{x}$ between data distributions where to approximate Coulomb field $E(\widetilde{x})$ at these points. In principle, we can use any other way to define the intermediate points.
>
> To illustrate the aforementioned fact, we conduct the following 2-dimensional experiment with Swiss roll dataset. In the first case, we define the inter-plate points via the interpolation $tx + (1-t)y$ and approximate Coulomb field there. In the second case, we define uniform cube mesh between plates. The result of experiments demonstrates that the performance does not depend on interpolation. Please, familiarize yourself with the Fig.5 via the main link.
>
> Since the geometry of data distributions is sufficiently difficult in high-dimensional spaces, we use Eq. (19) as the way to define intermediate points just to cover a bigger volume of space.
>
> **Concluding remarks.** We would be grateful if you could let us know if the explanations we gave are satisfactory. If so, we kindly ask that you consider increasing your rating. We are also open to discussing any other questions you may have.

---

### Official Review · Reviewer_aVB4 · 2025-03-15

**Overall Recommendation:** 3

**Summary:**

The paper introduces Electrostatic Field Matching (EFM), a novel generative modeling framework inspired by the physics of an electrical capacitor. In EFM, source and target data distributions are assigned positive and negative charges on two parallel plates, and a neural network is used to learn the resulting electrostatic field. By moving samples along the field lines from one plate to the other, the method provably transforms the source distribution into the target distribution. This approach is versatile, addressing both noise-to-data and data-to-data generation tasks, and its theoretical guarantees and experimental results on toy and image datasets position it as a compelling alternative to existing diffusion and flow-based models.

**Claims And Evidence:**

The method’s performance on high-dimensional or complex tasks (real-images) is not convincingly demonstrated, and its sensitivity to hyper-parameters—such as the inter-plate distance—raises concerns about practical robustness.

**Essential References Not Discussed:**

None

**Experimental Designs Or Analyses:**

The method is evaluated on three tasks:
1. Gaussian-to-Swiss Roll Experiment: We consider a 2-dimensional, zero-centered Gaussian distribution with an identity covariance matrix as P(x+), and a Swiss Roll distribution as Q(x−).
2. Image-to-Image Translation Experiment: This task involves transforming colored digit 3 into colored digit 2.
3. Image Generation Task: This involves generating 32×32 colored images of digit 2 from the MNIST dataset.

**Methods And Evaluation Criteria:**

This paper primarily generates visual results, while quantitative results are lacking.

**Other Comments Or Suggestions:**

I would like to raise rating if fair quantitive results and the evidence that this method can be scaled can be presented.

**Other Strengths And Weaknesses:**

1. The methods are well-motivated by electrostatic theory and the chosen evaluation tasks—using toy datasets and colored MNIST experiments—are standard for proof-of-concept generative modeling.

1. However, further testing on more diverse and complex datasets could strengthen the practical validation.

**Questions For Authors:**

None

**Relation To Broader Scientific Literature:**

Shaul, Neta, et al. "Flow Matching with General Discrete Paths: A Kinetic-Optimal Perspective." arXiv preprint arXiv:2412.03487 (2024).

**Theoretical Claims:**

Yes, I have checked definitions and theorems in 3.1.

---

> ### Author Rebuttal · Authors · 2025-04-01
>
> Dear reviewer, thank you for reviewing our paper. Below we answer to your questions and comments.
>
> **(Q1) [...] performance on high-dimensional or complex tasks (real-images) [...] testing on more diverse and
> complex datasets [...] quantitative results are lacking..**
>
> It is worth noting that our main goal is to demonstrate proof-of-concept of our method in the experimental section. We agree that providing additional generating experiments with more complex datasets enhances the understanding of method's performance. Further scalability of the method is a promising avenue for future research.
>
> Nevertheless following your request, we include **additional experiments** with more complex data such as CIFAR-10. For qualitative analysis, we demonstrate our EFM's results as well as PFGM's performance. We would like to ask you to get acquainted with Fig. 3 that is available via the anonymous link https://drive.google.com/file/d/1DTbQR_GNah7hVGGnDF822aD96iWxjR-k/view?usp=sharing.
>
> For quantitative analysis, we calculate FID and CMMD scores on test part of the aforementioned datasets and compare our method with PFGM, DDPM, DDIM and GLOW. Firstly, we demonstrate quantitative performance of our method on full Colored MNIST dataset. We see that our method outperforms other approaches on full Colored MNIST, reaching the lowest FID and CMMD.
>
> | Metrics/Method | EFM (our) | PFGM | DDPM | DDIM | GLOW |
> |----------------|----------|------|------|------|------|
> | FID            | **0.92** | 1.88 | 2.18 | 2.23 | 25.9 |
> | CMMD           | **1.47** | 2.28 | 2.68 | 2.85 | -    |
>
> Secondly, we demonstrate quantitative performance of our method on CIFAR-10 dataset. We see that in image generation on CIFAR-10, our performance is comparable to PFGM. *At the same time, we remind that our method is also capable of performing data-to-data transfer while PFGM is not capable of doing that*.
>
> | Metrics/Method | EFM (our) | PFGM       | DDPM | DDIM | GLOW  |
> |----------------|-----------|------------|------|------|-------|
> | FID            | 2.62      | **2.48**   | 3.17 | 4.16 | 48.9  |
> | CMMD           | **1.87**  | 1.93       | 2.98 | 3.25 | -     |
>
> Additionally, we quantitatively and qualitatively compare the performance of our approach with SB-based (DDIB ,$\alpha$-DSBM ) approaches as well as GAN-based (Cycle-GAN ) for unpaired data setups on Colored MNIST. Please, familiarize yourself with Fig.6 via the afore mentioned link. For quantitative analysis, we demonstrate FID and CMMD metrics on Colored MNIST dataset for our and compared methods.
>
> | Metrics/Method | EFM (our) | FM    | DSBM | DDIB | GAN  |
> |----------------|-----------|-------|------|------|------|
> | FID            | **4.45**  | 19.87 | 7.21 | 8.24 | 4.57 |
> | CMMD           | **2.37**  | -     | 4.02 | 4.11 | 2.45 |
>
> We see that our method demonstrates the highest FID and CMMD among compared approaches.
>
> **(Q2) [...] sensitivity to hyper-parameters  [...] The inter-plate distance**
>
> We conduct additional experiment that deals with the influence $L$ on the performance ( see Fig.4 via the link). The more distance between plates the worse approximation of the field. If $L$ is too small, field is recoverable , but there is "edge effect" and it might has influence on performance.
>
>
> **Concluding remarks.** We would be grateful if you could let us know if the explanations we gave are satisfactory. If so, we kindly ask that you consider increasing your rating. We are also open to discussing any other questions you may have.

---

### Official Review · Reviewer_d6Dp · 2025-03-22

**Overall Recommendation:** 3

**Summary:**

The paper proposes Electrostatic Field Matching (EFM), a method for generative modeling and distribution transfer based on electrostatic principles. EFM generalizes the Poisson Flow Generative Model (PFGM) by enabling mapping between arbitrary distributions. It represents source and target distributions as charged capacitor plates and learns the electrostatic field between them using a neural network.

**Claims And Evidence:**

The authors introduce a new generative approach and present proof-of-concept experiments. Their claims are supported by visual evaluations.

**Essential References Not Discussed:**

The references in this paper are relevant.

**Experimental Designs Or Analyses:**

Lack of reproducibility: The code in the Appendix imports functions that are not included in the provided package.

The paper is presented as a proof-of-concept without quantitative experiments, providing only visual results of the method. The details of the visual experiments are given in Appendix C and source code.

**Methods And Evaluation Criteria:**

The benchmarks are relevant but very limited. Only toy datasets (Swiss Roll and MNIST) are used, with no quantitative evaluations or comparisons to other methods from the literature.

**Other Comments Or Suggestions:**

I wonder if, in Algorithm 1, t should be sampled from Uniform(0, 1) to ensure proper interpolation between x+ and x-.

**Other Strengths And Weaknesses:**

Strengths:
- The authors propose a creative method that uses Poisson Flow [1] as a Schrödinger Bridge for interpolating between two arbitrary distributions.
- The authors claim that the method works with unpaired datasets.
- The paper is well-written and easy to follow, with the Related Works section being particularly well-structured.


Weaknesses:
- There are no quantitative evaluations of the method; the authors provide only visual results.
- The method is not compared to alternative approaches from the literature (e.g., SB-based: [2, 3, 4], GAN-based: [5]).
- The authors use only toy datasets (Swiss Roll, MNIST). Using more complex datasets (like CIFAR or ImageNet) would help determine whether the estimated field is sensitive to the dimensionality of the data.
- In line 250, the authors mention that the inference process is summarized in Algorithm 1, but the algorithm only describes the training procedure.
- The source code utilizes functions not included in the provided package, making it impossible to reproduce the method.

Weaknesses 1-3 limit the work to a proof-of-concept, reducing its overall contribution.

[1] Xu, Yilun, et al. "Poisson flow generative models." Advances in Neural Information Processing Systems 35 (2022): 16782-16795.
[2] Su, Xuan, et al. "Dual diffusion implicit bridges for image-to-image translation." arXiv preprint arXiv:2203.08382 (2022).
[3] Kim, Beomsu, et al. "Unpaired image-to-image translation via neural schr\" odinger bridge." arXiv preprint arXiv:2305.15086 (2023).
[4] De Bortoli, Valentin, et al. "Schrodinger Bridge Flow for Unpaired Data Translation." Advances in Neural Information Processing Systems 37 (2024): 103384-103441.
[5] Zhu, Jun-Yan, et al. "Unpaired image-to-image translation using cycle-consistent adversarial networks." Proceedings of the IEEE international conference on computer vision. 2017.

**Questions For Authors:**

- As far as I understand, the field is estimated based on all samples from the batch. How does the batch size influence the stability of the training? Is it possible to train such a model with the batch size of 1?
- Unlike PFGM, your approach does not project the Q distribution on the (D+1)-dimensional hemisphere, but places it on z=L hyperplane. I wonder if samples that are on the periphery of the P distribution would not be pushed away and land far away from Q distribution as a result?
- The hyperparameter L seems crucial, but it is difficult to tune. Based on the experiments, if the data has higher dimensionality, L should be increased. Is it possible to propose a function that assigns an L value for any given number of dimensions D?

**Relation To Broader Scientific Literature:**

The proposed method extends the idea introduced in [1] by replacing the source distribution—from a known uniform distribution projected onto a (D+1)-dimensional hemisphere—with any arbitrary distribution placed on a hyperplane parallel to the target distribution.

This concept of constructing the vector field from one distribution to another is known as a Schrodinger Bridge in Diffusion Models nomenclature.

[1] Xu, Yilun, et al. "Poisson flow generative models." Advances in Neural Information Processing Systems 35 (2022): 16782-16795.

**Theoretical Claims:**

I did not find any issues with the theoretical claims in the paper. The manuscript primarily follows the derivations from [1].


[1] Xu, Yilun, et al. "Poisson flow generative models." Advances in Neural Information Processing Systems 35 (2022): 16782-16795.

---

> ### Author Rebuttal · Authors · 2025-04-01
>
> We thank the reviewer for the valuable comments. Please find the answers to your questions below.
>
> **(Q1) [...] quantitative evaluations or comparisons [...] alternative approaches from the literature (e.g., SB-based, GAN-based). CIFAR [...]**
>
> Following your request, we include **additional experiments** with more complex data such as CIFAR-10. For qualitative analysis, we demonstrate our EFM's results as well as PFGM's performance. See Fig.3 via the anonymous link https://drive.google.com/file/d/1DTbQR_GNah7hVGGnDF822aD96iWxjR-k/view?usp=sharing. For quantitative evaluation,  we calculate FID and CMMD[1] on test of CIFAR-10 and compare with PFGM, DDPM, DDIM and GLOW. Please see the table with results in the [answer to YMMk](https://drive.google.com/file/d/1DTbQR_GNah7hVGGnDF822aD96iWxjR-k).We see that in image generation our performance is comparable to PFGM. *At the same time, we remind that our method is also capable of performing data-to-data transfer while PFGM is not capable of doing that*.
>
> In accordance of your request of comparison with GAN and SB-based methods, we also demonstrate the performance of our approach with SB-based (DDIB,$\alpha$-DSBM ) approaches as well as GAN-based (Cycle-GAN ) for unpaired data setups on Colored MNIST. Please, familiarize yourself with Fig.6 via the afore mentioned link. We also see that our method demonstrates the highest FID and CMMD among compared approaches.
>
> | Metrics/Method | EFM (our) | FM    | DSBM | DDIB | GAN  |
> |----------------|-----------|-------|------|------|------|
> | FID            | **4.45**  | 19.87 | 7.21 | 8.24 | 4.57 |
> | CMMD           | **2.37**  | -     | 4.02 | 4.11 | 2.45 |
>
> **(Q2) The manuscript primarily follows the derivations from [1].**
>
> We respectfully disagree. Our significant theoretical advancement compared to PFGM is using the following fundamental property of electric field lines in $D$-dimensional space: field lines starting from the positive charge distribution $\mathbb{P}(\cdot)$ almost surely terminate in the negative charge distribution $\mathbb{Q}(\cdot)$ (Lemma A.7). This property — *previously not considered in electrostatic generative models* — combined with field flux conservation along current tubes, formally establishes (our Theorem 3.1) that field line trajectories transport samples between $\mathbb{P}$ and $\mathbb{Q}$. This constitutes our main theoretical contribution enabling data-to-data transfer.
>
> **(Q3) In line 250, the authors mention that the inference process is summarized in Algorithm 1, but the algorithm only describes the training procedure.**
>
> Thanks for pointing that out. The inference algorithm is just a simulation of the learned ODE as described in Section 3. For convenience of the reader, we will add a separate algorithm box with inference procedure in the final version of the paper.
>
> **(Q4) I wonder if, in Algorithm 1, t should be sampled from Uniform(0, 1) to ensure proper interpolation between x+ and x-**
>
> The uniform sampling $t \sim \text{Uniform}(0,1)$ represents no more than a particular training volume selection strategy for interpolating between $\widetilde{\mathbf{x}}^+$ and $\widetilde{\mathbf{x}}^- $, controlling the density of training points along the interplanar axis. As demonstrated in Fig. 5 (link from Q1) for the Gauss-to-Swiss-roll experiment, both the proposed uniform training volume (Algorithm~1) and conventional cubic lattice initialization produce nearly identical electric field line configurations.
>
> **(Q5) How does the batch size influence the stability of the training? [...] batch size of 1?**
>
> Theoretically, it is possible to train our method with a batch size equal to 1. However, Monte Carlo integration has a higher variance for a small number of samples. Following the question, we conducted an extra experiment with different batch sizes. We demonstrate that performance almost stops to increase with batch sizes exceeding 64, see Fig. 2 in extra material (link from Q1)
>
> **(Q6) I wonder if samples that are on the periphery of the P distribution would not be pushed away and land far away from Q distribution as a result?**
>
>  This scenario is precluded by Theorem 3.1, which guarantees transport between distributions $\mathbb{P}(\cdot)$ and $\mathbb{Q}(\cdot)$. While peripheral samples exhibit greater curvature compared to central trajectories due to boundary effects, this geometric difference does not prevent their convergence to $\mathbb{Q}(\cdot)$. The continuity of field lines (Lemma A.7) ensures termination on the target distribution almost surely.
>
> **(Q7) [...] source code [...] unable to run. [...] functions not included in the provided package**
>
> We will fix this in the final code version.
>
> **Concluding remarks.** We would be grateful if you could let us know if the explanations we gave are satisfactory. If so, we kindly ask that you consider increasing your rating. We are also open to discussing any other questions you may have.

---

> > ### Comment · Reviewer_d6Dp · 2025-04-03
> >
> > Thank you for answering my questions.
> >
> > **(Q1) [...] quantitative evaluations or comparisons [...] alternative approaches from the literature (e.g., SB-based, GAN-based). CIFAR [...]**
> >
> > The experiments you conducted are sufficient and effectively demonstrate how your method works. It enhances the value of your paper significantly. I kindly suggest including these tables and figures in the final version of the manuscript.
> >
> > I have one additional question regarding the backbone used in these experiments. Specifically, do the different approaches share the same neural network architecture and have the same number of function evaluations (NFE) during inference?
> > If they do, it would be worth highlighting. Otherwise, I kindly suggest adding two rows to the results table: one for the number of parameters and another for the number of network evaluations.
> >
> > **(Q2) The manuscript primarily follows the derivations from [1].**
> >
> > I was pointing out that your work has a similar theoretical background and, therefore, shares certain derivations. My intention was not to undermine the contribution of your paper.
> >
> > **(Q4) I wonder if, in Algorithm 1, t should be sampled from Uniform(0, 1) to ensure proper interpolation between x+ and x-**
> >
> > I have to admit that this part is the most confusing to me.
> >
> > From line 257, we know that $\tilde{x}^+ = [x^+, 0]$ and $\tilde{x}^- = [x^-, L]$, which positions the two distributions on parallel planes with a distance of $L$.
> >
> > If we sample $t \sim Uniform(0, L)$ and $L > 1$, then following the formula $$x_t = t \tilde{x}^+ + (1 - t) \tilde{x}^- + \epsilon. $$
> >
> > could lead to **linear extrapolation** rather than interpolation.
> >
> > As a result, this might produce negative values in the time dimension, and the extrapolation could significantly increase the variance of the prior distribution. Was this intentional?
> >
> > **(Q5) How does the batch size influence the stability of the training? [...] batch size of 1?**
> >
> > I’m glad this experiment was conducted, as I find it especially valuable for the community.
> >
> > At a high level, we can say that SB methods trained on paired datasets (e.g., I2SB, ResShift, InDI, IR-SDE, etc.) use one positive and one negative sample to construct the ground truth vector field. This approach works well for paired tasks such as image enhancement.
> > However, in domain shift scenarios where the data is unpaired, the results tend to concentrate in regions where the digits appear very bold. I believe this represents plausible mass centers of clusters (medoids) within the target distribution, likely due to averaging in the vector field. When using a larger batch size, this issue disappears, as the approximations become more precise.
> >
> > **(Q6) I wonder if samples that are on the periphery of the P distribution would not be pushed away and land far away from Q distribution as a result?**
> >
> > Your arguments are valid when discussing the theoretical vector field.
> > However, the goal here is to train a neural network to approximate that field. This approximation may be biased due to: (a) Monte Carlo sampling used to estimate the vector field during training, and (b) inherent imperfections in the model.
> >
> > My main concern is that when a sample lies at the boundary of the prior distribution, the network is trained to **push** it even further. This could create issues in later stages of the trajectory, as the space expands and the model must learn the vector field over a much broader region. This approach contrasts with most SB methods, which aim to find an optimal transport path, thereby reducing the intermediate space that needs to be modeled (eg. https://arxiv.org/pdf/2302.00482).
> >
> >
> > **Conclusion**
> > The additional experiments have significantly enhanced the value of this submission. As a result, I believe my initial score is no longer appropriate, and I would like to raise it after I receive short answers to **Q4** and **Q6**.

---

> > > ### Author Response · Authors · 2025-04-05
> > >
> > > Dear reviewer, we are exceedingly glad that you found our additional experiments as the efficient demonstration of our methodology, pointing out the significance of our results. Undoubtedly, we are going to add new additional experiments, quantitative evaluations and comparisons in the final version of our paper.
> > >
> > > **(Q1) [...] the same neural network architecture and have the same number of function evaluations (NFE) during inference?[...]**
> > >
> > > Undoubtedly, we use the same neural network architecture for PFGM, DDPM and our approach to provide honest comparison. Also, the majority of hyper-parameters (learning rate, ema decay and so on) are the same as in PFGM because our codebase is based on PFGM code. As for NFE, we also use the same 100 steps for inference process for our EFM as well as PFGM, following PFGM's configs of code.
> > >
> > >
> > > **(Q4) [...]   I wonder if, in Algorithm 1, $t$ should be sampled from Uniform(0, 1) to ensure proper interpolation between $x+$ and $x-$ [...]**
> > >
> > > We always sample $t$ from uniform distribution Uniform(0,1).
> > > There is the typo in the text and we will correct this moment in the final version of our paper. We are grateful to you for pointing it out.
> > >
> > > **(Q6) [...]  I wonder if samples that are on the periphery of the P distribution would not be pushed away and land far away from Q distribution as a result?  [...]**
> > >
> > > Peripheral samples generate field lines that exhibit greater curvature compared to central trajectories. While these curved paths originate from positive charges, they necessarily terminate on negative charges. Thus, the network will not push the peripheral samples further away in our framework.
> > >
> > > The only real problem with peripheral samples is the problem of limited training volume. When field lines extend beyond this volume, reliable transport cannot be guaranteed. Strictly speaking, complete coverage would require training across an infinite volume. However, since the electric field strength decays as $1/r^{D-1}$, in practice this ensures that most of the field lines remain within the finite region.
> > >
> > > Furthermore, appropriate selection of the hyper parameter $L$ can render a significant portion of field lines nearly straight (see the toy experiment in Figure 7). However, optimal training volume selection remains an open research question.
> > >
> > > Regarding Monte Carlo sampling, while batch size equal to 1 is theoretically sufficient, we observe an improved convergence with larger batches (Figure 2 in supplementary materials), with performance saturating at batch size = 64.

---

### Official Review · Reviewer_cYJh · 2025-03-23

**Overall Recommendation:** 3

**Summary:**

In this work, the authors propose Electrostatic Field Matching (EFM), which transforms between two distributions in the same space by placing the two distributions on two parallel plates with opposite charge, training a neural network to predict the electric field in the space between the two plates, and trace a test charge through the field from one plate to the other. Beside theoretical justifications, some interesting results on random generation and translation between different digits on a colored MNIST dataset are shown as a proof of concept.

**Claims And Evidence:**

While the theoretical proofs for the principles appear sound, the experimental results are limited, as acknowledged by the authors who present them as toy examples without making broad claims.

**Essential References Not Discussed:**

No

**Experimental Designs Or Analyses:**

Experiments on more complex datasets are desirable. It seems to me that the jump from “placing the target distribution on a changed plate surrounded by an infinite sphere and run a test charge” (PFGM) to “placing two distributions on two parallel plates with opposite charges and run a test charge” (proposed method) is not a big one, especially considering that the underlying theory is the same, so I would say this concept is in large parts already proven. Thus, I think mere proof-of-concept experiments are not sufficient. Even if we are only getting simple examples, there should at least be the result of random generation on the full color-MNIST dataset, instead of just a subset featuring the same digit.

In addition, some quantitative evaluations would also help.

**Methods And Evaluation Criteria:**

I have several concerns about the method. First, how can we ensure accurate approximation of the ground truth field using Monte Carlo integration? In high dimensions, field contributions from distant charges decay rapidly with distance, making nearby charges significantly more important. However, in high dimensions, Monte Carlo samples are unlikely to land near these crucial regions, potentially leading to poor sampling of the most important areas. Though I acknowledge PFGM faces a similar challenge.

Second, parallel plates with opposite charges create a dipole effect, where field lines inevitably spread into the surrounding space rather than staying confined between the plates. Some field lines initially travel backward, away from the target plate, make extensive detours, and eventually reach the target plate from behind. While PFGM can focus on one side due to symmetry, the lack of symmetry here raises questions about whether we can safely consider only the "right" side while ignoring the "wrong" side. Even if we can focus on one side, field lines can travel far from the plates before returning—a significant issue in high dimensions where most directions are approximately perpendicular. When training samples are created by interpolating between two random points on the plates, they remain between the plates. This raises the question: how can we ensure accurate tracing of outward field lines? PFGM avoids this issue since it intentionally allows field lines to extend to infinity.

Finally, there's the matter of parameter L. The Swiss Roll experiment demonstrates L's crucial role—larger values cause more field lines to deviate sideways, negatively affecting results. While smaller L values likely have their own drawbacks, the authors should provide guidelines for determining appropriate L values.

**Other Comments Or Suggestions:**

The authors dedicate one and a half pages to basic physics concepts that, given their elementary nature indicated by the section heading, could be condensed. This space would be better used for additional experimental results.

**Other Strengths And Weaknesses:**

No

**Questions For Authors:**

1.What is the effect if L is too small?
2. Is it safe to ignore field lines that start out facing the wrong direction?

**Relation To Broader Scientific Literature:**

Not evaluated

**Theoretical Claims:**

I didn’t find any specific issues.

---

> ### Author Rebuttal · Authors · 2025-04-01
>
> Dear reviewer, thank you for reviewing our paper. Below we answer your questions.
>
> **(Q1) [...] Experiments on more complex datasets[..]some quantitative evaluations would also help.**
>
> Following your request, we include **additional experiments** with more complex data such as CIFAR-10 and the requested full color-MNIST dataset. For qualitative analysis, we demonstrate our EFM's results as well as PFGM's performance. We ask you to get acquainted with Fig. 1 and Fig. 3 that are available via the anonymous link  https://drive.google.com/file/d/1DTbQR_GNah7hVGGnDF822aD96iWxjR-k/view?usp=sharing. For quantitative analysis, we calculate FID/CMMD scores and compare with another methods. Please, familiarize yourself with the the [answer to YMMk](https://openreview.net/forum?id=9dHilxylvC&noteId=1boFlYDdKN).
>
> **(Q2) [...] accurate approximation of the  field using Monte Carlo integration? [...] In high dimensions, field contributions from distant charges decay rapidly with distance [...]**
>
> If we correctly understand your question, your ask about the accurate approximation of $E(\widetilde{x})$ in Eq. (11) for a given $\widetilde{x}$ via Monte Carlo sampling. Since Monte Carlo integration provides an unbiased estimation of an integral and its variance does not depend on the dimensionality, this estimation might be used for accurate approximation of the field. However, the more samples are used in batch for estimation the lower variance. We conduct **additional experiments** with different batch sizes and its influence on our EFM's performance. Please, familiarize yourself with the Fig. 2 via the main link.
>
> **(Q3) [...] dipole effect [...] Some field lines initially travel backward [...] reach the target plate from behind. [...] how can we ensure accurate tracing of outward field lines?**
>
> Indeed, each plate emits two distinct sets of field lines. One set is directed towards the second plate and another oriented in the opposite direction. Crucially, the properties of electric field lines (LemmaA.7) ensure that both sets almost surely terminate on the opposing plate. The primary distinction lies in their geometric trajectories: the forward-directed lines exhibit smaller curvature and reach the target plate more efficiently (faster) than their backward-oriented counterparts. From a practical point of view, prioritizing forward-directed trajectories is advantageous for computational efficiency while remaining theoretically sound due to Theorem~3.1, which guarantees almost sure transport from $\mathbb{P}(\cdot)$ to $\mathbb{Q}(\cdot)$.
>
> To address field lines extending beyond the plate boundary ($z > L$) before reaching $\mathbb{Q}(\widetilde{\mathbf{x}}^-)$, one may use the following natural criterion to distinguish the valid termination on $\mathbb{Q}(\cdot)$ from the transient crossings of $z = L$:
>
> \begin{equation}
>     \begin{cases}
>         E_z(z \to L^-) = E_z(z \to L^+) & \implies \text{The line goes away past the distribution}, \\
>         E_z(z \to L^-) \neq E_z(z \to L^+) & \implies \text{Valid termination on } \mathbb{Q}(\widetilde{\mathbf{x}}^-).
>     \end{cases}
> \end{equation}
>
> This criterion derives from:
>  1) Field continuity along current tubes (Lemma A.3, Corollary A.4),
>  2) Boundary field behavior (LemmaA.5): Near a plate, the field is determined by the charge of this plate and is directed away from the plate in the case of positive charge (and towards the plate in the case of negative charge, see lemma A.5). Therefore the direction and magnitude of the field must change in the case if we arrive at the target distribution.
>
> Thus, discontinuities in $E_z$ explicitly signal successful transitions to $\mathbb{Q}(\cdot)$, while continuity indicates a need for further integration.
>
> **(Q4) matter of parameter L [...]**
>
> We conduct **additional experiment** that deals with the influence $L$ on the performance (see Fig.4 via the link). The more distance between plates the worse approximation of the field. If L is too small, field is recoverable , but there is "edge effect" and it might has influence on performance.
>
> **(Q5) PFGM [...] the underlying theory is the same [...]**
>
> We respectfully disagree. Our significant theoretical advancement compared to PFGM is using the following fundamental property of electric field lines in $D$-dimensional space. The field lines starting from the positive charge distribution $\mathbb{P}(\cdot)$ almost surely terminate in the negative charge distribution $\mathbb{Q}(\cdot)$ (Lemma A.7). This property — *previously not considered in electrostatic generative models* — combined with field flux conservation along current tubes, formally establishes (our Theorem 3.1) that field line trajectories transport samples between $\mathbb{P}$ and $\mathbb{Q}$. This constitutes our main theoretical contribution.
>
> **(Q6) Is it safe to ignore field lines that start out facing the wrong direction?**
>
> Yes, it is safe due to our main theorem. See the answer to the **Q3** for details.

---

### Decision · Program_Chairs · 2025-05-01

**Decision:**

Accept (poster)

**Comment:**

This paper presents an interesting variation of modeling the electrostatic interaction between two distributions by placing them on parallel plates with opposite charges. The conceptual idea is creative and well-motivated, drawing inspiration from physical analogies in a meaningful way.

One reviewer raised concerns regarding the novelty of the approach, noting similarities with rectified flows. However, the authors provided a satisfactory explanation during the rebuttal phase, clarifying the distinctions and contributing to a better understanding of their approach.

The primary weakness lies in the empirical evaluation. The initial results were limited to relatively simple benchmarks, which made it difficult to fully assess the method's effectiveness. The authors addressed this by providing more comprehensive results during the rebuttal stage. While these additions improved the evaluation, the method has not yet been demonstrated on high-dimensional datasets, which limits the scope of its practical applicability.

In conclusion, while the idea is novel and the theoretical framing is compelling, the empirical validation remains somewhat limited. Therefore, I am marginally in favor of accepting the paper, acknowledging its potential and the constructive efforts made during the rebuttal.